**Effects of pH and light exposure on the survival of bacteria and their ability to biodegrade**
**organic compounds in clouds: Implications for microbial activity in acidic cloud water**
Yushuo Liu,[1,2] Chee Kent Lim,[1] Zhiyong Shen,[1] Patrick K. H. Lee,[1,3] Theodora Nah[1,2,3*]
*[1]School of Energy and Environment, City University of Hong Kong, Hong Kong SAR, China*
*[2]Shenzhen Research Institute, Nanshan District, Shenzhen, China*
*[3]State Key Laboratory of Marine Pollution, City University of Hong Kong, Hong Kong SAR, China*
*\* To whom correspondence should be addressed: Theodora Nah (Email: theodora.nah@cityu.edu.hk)*
**Abstract**
Recent studies have reported that interactions between live bacteria and organic matter can
potentially affect the carbon budget in clouds, which has important atmospheric and climate
implications. However, bacteria in clouds are subject to a variety of atmospheric stressors,
which can adversely affect their survival and energetic metabolism, and consequently their
ability to biodegrade organic compounds. At present, the effects of cloud water pH and solar
radiation on bacteria are not well understood. In this study, we investigated how cloud water
pH (pH 3 to 6) and exposure to solar radiation impact the survival and energetic metabolism
of two *Enterobacter* bacterial strains that were isolated from ambient air collected in Hong
Kong and their ability to biodegrade organic acids. Experiments were conducted using
simulated sunlight (wavelength 320 to 700 nm) and microcosms comprised of artificial cloud
water that mimicked the pH and chemical composition of cloud water in Hong Kong, South
China. Our results showed that the energetic metabolism and survival of both strains depended
on the pH. Low survival rates were observed for both strains at pH < 4 regardless whether the
strains were exposed to simulated sunlight. At pH 4 to 5, the energetic metabolism and survival
of both strains were negatively impacted only when they were exposed to simulated sunlight.
Organic compounds such as lipids and peptides were detected during exposure to simulated
sunlight at pH 4 to 5. In contrast, there were minimal effects on the energetic metabolism and
survival of both strains when they were exposed to simulated sunlight at pH > 5. The
biodegradation of organic acids was found to depend on the presence (or absence) of simulated
sunlight and the pH of the artificial cloud water medium. Overall, this study provides new
insights into how two common atmospheric stressors, cloud water pH and exposure to solar
radiation, can influence the survival and energetic metabolism of bacteria, and consequently
the roles that they play in cloud processes.

## 1. Introduction

Clouds are an important medium for the aqueous-phase formation and transformation of organic and inorganic compounds. In addition to inorganic and organic compounds, clouds contain biological matter including biological debris (e.g., dead cells, cell fragments) and live microorganisms (e.g., bacteria, fungal spores) (Bauer et al., 2002; Jaenicke, 2005; Burrows et al., 2009). Live microorganisms are mainly emitted directly into the atmosphere from natural sources (Jaenicke, 2005; Möhler et al., 2007; Burrows et al., 2009; Attard et al., 2012; Hu et al., 2018). Once airborne, they can participate in a variety of atmospheric processes such as cloud formation, precipitation, ice nucleation, and the microbial degradation of atmospheric organics (Amato et al., 2005; Delort et al., 2010; Vaitilingom et al., 2010; Vaitilingom et al., 2013; Morris et al., 2014; Morris et al., 2017; Hu et al., 2018; Huang et al., 2021; Zhang et al., 2021). Bacteria are incorporated into clouds through nucleation and scavenging processes (Möhler et al., 2007). So far, only bacterial communities in clouds in some areas (e.g., Puy de Dôme in France, Mt. Tai in North China) have been extensively investigated. These studies showed that the bacterial communities in clouds are highly complex and diverse, and mainly originate from vegetation, soil, and water bodies (Vaïtilingom et al., 2012; Wei et al., 2017; Zhu et al., 2018). A significant fraction of the bacteria in clouds may be major allergens and/or pathogens that originate mainly from anthropogenic activities, and their concentrations usually increase during air pollution episodes (Wei et al., 2017; Peng et al., 2019). The cell concentrations of bacteria in clouds typically range from about $10^2$ to $10^5$ cells mL$^{-1}$ (Amato et al., 2005; Burrows et al., 2009; Amato et al., 2017). At present, our knowledge on bacterial communities in clouds are limited to the few areas that have been studied (e.g., Puy de Dôme in France, Mt. Tai in North China) (Amato et al., 2005; Amato et al., 2017; Wei et al., 2017; Péguilhan et al., 2021). Cultural bacteria typically makes up a very small fraction (about 1%) of the entire bacteria community in clouds (Amato et al., 2005).

Airborne bacteria are comprised of both dead or dormant cells and metabolically active cells. Previous culture-based and culture-independent analyses of bacteria isolated from cloud water have shown that some of these bacteria species are metabolically active (Amato et al., 2007; Krumins et al., 2014; Amato et al., 2019). Previous studies have reported that the

degradation of organic compounds as a result of microbiological-chemical interactions between live bacteria and organic matter can play an important role in influencing the carbon budget in clouds, which will have important atmospheric and climate implications (Delort et al., 2010; Vaitilingom et al., 2010; Vaitilingom et al., 2013; Ervens and Amato, 2020). Many bacteria species have the enzymes needed to biodegrade organic compounds. Some of the bacteria species isolated from cloud water could biodegrade organic acids, formaldehyde, methanol, phenolic compounds, and amino acids (Ariya et al., 2002; Husárová et al., 2011; Vaïtilingom et al., 2011; Jaber et al., 2020; Jaber et al., 2021). However, the bacteria are exposed to a variety of stressors that can negatively impact their survival and microbial activity in clouds. Joly et al. (2015) previously investigated the individual impacts of osmotic shocks, freeze-thaw cycles, and exposure to light and $H_2O_2$ on the survival of different bacterial strains in microcosms mimicking cloud water chemical composition at Puy de Dôme. Osmotic shocks and freeze-thaw cycles reportedly had the greatest negative impacts on the survival of bacteria, while exposure to light and $H_2O_2$ had limited impacts on the survival of bacteria. However, there are other stressors that bacteria in clouds are commonly subjected to beyond the four stressors investigated by Joly et al. (2015). In addition, when combined together, the stressors may have synergistic negative impacts on the survival and microbial activity of bacteria in clouds. The potentially synergistic negative impacts that stressors have on the survival and microbial activity of bacteria in clouds have yet to be investigated. Some bacteria species respond to stressors by releasing organic compounds (e.g., proteins, pigments, lipids) as a defensive mechanism (Davey and O'toole, 2000; Delort et al., 2010; Flemming and Wingender, 2010; Vaïtilingom et al., 2012; Matulova et al., 2014). When bacteria species cannot withstand the stress, the resulting cellular damage and lysis will lead to the release of biological material. In addition, the ability of bacteria to biodegrade organic compounds in clouds will decrease if their metabolism and survival are negatively impacted.

Cloud water acidity is another stressor that bacteria are subjected to in clouds. There has been limited study on the impact of cloud water pH on the survival and microbial activity of bacteria in clouds. However, some studies have reported that the cloud water pH impacts the diversity and composition of bacterial communities (Amato et al., 2005; Peng et al., 2019).

For instance, spore-forming bacteria were abundant in pH 4.9 cloud water at Puy de Dôme, while more diverse and higher concentrations of non-spore-forming bacteria were observed in pH 5.8 cloud water (Amato et al., 2005). The pH of cloud water typically lies between 3 and 6 (Pye et al., 2020), with a global mean of around pH 5.2 (Shah et al., 2020). Areas with high inputs of sulfuric acid and/or nitric acid combined with low inputs of ammonia, dust, and sea salt, especially in parts of East Asia, have moderately acidic to highly acidic cloud water (pH < 5) (Li et al., 2020; Pye et al., 2020; Shah et al., 2020; Qu and Han, 2021). To the best of our knowledge, there has been no studies on how moderately acidic to highly acidic cloud water affects the survival and microbial activity of bacteria. The effects of light exposure on the survival and microbial activity of bacteria are also ambiguous. Some studies reported that exposure to UVA and visible light will lead to the formation of intracellular reactive oxidative species, which can damage important cell components and cause cell death (Anglada et al., 2015). However, exposure to light reportedly did not impact the survival rates of bacterial strains from *Pseudomonas syringae*, *Arthrobacter* sp., and *Sphingomonas* sp. (Joly et al., 2015). While it is possible that exposure to acidic cloud water and light have a synergistic effect on the survival and microbial activity of bacteria, previous laboratory investigations were mainly performed in microcosms with the pH set between 5 to 7 to mimic cloud water in areas that have high inputs of ammonia, dust, and sea salt, such as the Puy de Dôme (Vaïtilingom et al., 2011; Joly et al., 2015; Jaber et al., 2021; Jaber et al., 2020).

This study investigates how cloud water pH and exposure to solar radiation affect the survival and energetic metabolism of bacteria and their ability to biodegrade organic compounds in clouds. We designed a series of laboratory experiments in microcosms containing artificial cloud water that mimicked the pH and chemical composition of atmospheric cloud water collected at the Tai Mo Shan station in Hong Kong, South China. South China is a region with moderately acidic to highly acidic cloud water due to its higher concentrations of acidic ions (e.g., $SO_4^{2-}$, $NO_3^-$) compared to alkaline ions (e.g., $NH_4^+$, $Ca^{2+}$) (Li et al., 2020; Qu and Han, 2021). Different pH (pH 3.3 to 5.9) and irradiation (illuminated vs. dark) conditions were employed in the experiments, during which we analyzed the biological material and organic compounds in the artificial cloud water medium at different

reaction time points. Since cloud water bacterial isolates from the Tai Mo Shan station are not available, two *Enterobacter* bacterial strains that were isolated from ambient air in Hong Kong were used as model bacteria in this study. In general, our current knowledge of the diversity and composition of bacteria communities in cloud water in Hong Kong and South China is very limited due to the scarcity of characterization studies conducted in this region. Results from a previous study reported that *Enterobacter* was one of the bacteria species in cloud water collected at the Nanling Mountain station in South China (Peng et al., 2019). *Enterobacter* bacteria has been detected in urban aerosols in different parts of the world, including South China (Chen et al., 2012; Després et al., 2012; Ding et al., 2015; Zhou et al., 2018; Prokof'eva et al., 2021). In addition, the enrichment of *Enterobacter* bacteria in the atmosphere during air pollution episodes has been reported in parts of Asia, America, and Europe (Romano et al., 2019; Ruiz-Gil et al., 2020; Romano et al., 2021). Since organic acids are ubiquitous in clouds (Tsai and Kuo, 2013; Löflund et al., 2002; Sun et al., 2016; Li et al., 2020) and can be biodegraded by most bacteria (Vaitilingom et al., 2010; Vaïtilingom et al., 2011), we chose seven organic acids that are commonly detected in clouds (formic acid, acetic acid, oxalic acid, maleic acid, malonic acid, glutaric acid, and methanesulfonic acid) as model organic compounds for our investigations of how cloud water pH and light exposure affect the ability of bacteria to biodegrade organic compounds in clouds.

**2. Methods**

**2.1. Strain isolation and whole genome sequencing**

Two new strains (B0910 and pf0910) belonging to *Enterobacter* species were isolated by exposing nutrient agar plates to ambient air in an urban environment (22.3360° N, 114.1732° E) at a height of 50 m above sea level during the summer season (~22 °C) in Hong Kong. The genomes of the two strains were sequenced using a GridION sequencer (Oxford Nanopore Technologies) by following the manufacturer's workflow. Genome assembly and the downstream genomic analyses are described in detail in Section S1. Based on genome comparison, *E. hormaechei* B0910 is most similar to *Enterobacter hormaechei* subsp. *hoffmannii* DSM 14563 (Average Nucleotide Identity (ANI) = 98.92) and *E. hormaechei*

pf0910 to *Enterobacter hormaechei* subsp. *steigerwaltii* DSM 16691 (ANI = 98.73) (Figure
S1). *E. hormaechei* B0910 has a chromosome (4.69 Mbp) with 4875 coding sequences (CDSs)
and a single plasmid (373 Kbp) with 383 CDSs. *E. hormaechei* pf0910 strain has a chromosome
(4.78 Mbp) with 5072 CDSs and two plasmids of 281 Kbp (344 CDSs) and 73 Kbp (79 CDSs).
**2.2. General experimental approach**

To simulate cloud water conditions in Hong Kong, artificial cloud water containing

major organic and inorganic ions in cloud water previously collected at the Tai Mo Shan station
(TMS; 22°24'N, 114°16'E, 957 m a.s.l.) were used in each experiment. Organic (acetic acid,
formic acid, oxalic acid, pyruvic acid) and inorganic (magnesium chloride, calcium chloride,
potassium chloride, sodium chloride, ammonium sulfate, ammonium nitrate, sodium hydroxide
and hydrochloric acid) compounds were used to prepare the artificial cloud water. Experiments
were performed using a Rayonet photoreactor (RPR-200, Southern New England Ultraviolet
Company). We followed the method employed in previous studies (George et al., 2015; Huang
et al., 2018; Misovich et al., 2021; Li et al., 2022) and used eight lamps with outputs centered
at different wavelengths to roughly simulate the range of solar radiation wavelengths (320 to
700 nm) inside the photoreactor. Figure S2 shows the resulting photon flux inside the
photoreactor. The temperature (25 °C) during the experiment was regulated by a fan located at
the bottom of the photoreactor.

The two strains were grown in LB broth at 37 °C to stationary phase. The culture was

then centrifuged at 6000 rpm for 10 min at 4 °C and the cell pellets were rinsed with artificial
cloud water (Table S1) three times. For investigations of the time evolution in the survival and
energetic metabolism of bacteria at different pH under illuminated vs. dark conditions (Section
2.2), the cells were re-suspended in artificial cloud water to an initial concentration of ~$10^5$
cells mL$^{-1}$. For investigations of the biodegradation of organic acids by bacteria at different pH
under illuminated vs. dark conditions (Section 2.3), the cells were re-suspended in artificial
cloud water to an initial concentration of ~$10^6$ cells mL$^{-1}$. A calibration curve was used to
convert between optical density and bacterial cell concentration.
Quartz tubes containing bacterial cells suspended in artificial cloud water (5 mL) were
placed on a rotating vial rack in the middle of the photoreactor. The quartz tubes for the dark
control experiments were wrapped in aluminum foil and placed inside the photoreactor. The
pH of the artificial cloud water did not change significantly during the experiments. Aliquots
of the solutions were taken at every hour over 12 hours for various offline chemical analyses.
100 µL of sample were removed at different time points for Colony Forming Unit (CFU) counts
on LB agar at 37 °C for 16 hours to determine the culturable bacterial cell concentrations, which
was used to calculate the bacteria survival rates. 20 µL of sample were removed at different
time points for measurements of adenosine diphosphate/adenosine triphosphate (ADP/ATP)
ratios using an assay kit (EnzyLight[TM], BioAssay Systems) and a biolumineter (SpectraMax
M2e) to determine changes in the bacteria energetic metabolism. All the experiments and
measurements were performed in triplicates.
**2.3. Investigations of the survival and energetic metabolism of bacteria at different pH**
**under illuminated vs. dark conditions**
Six pH conditions (pH 3.3, 4.3, 4.5, 4.7, 5.2 and 5.9) were chosen for this set of
experiments, which were performed under both dark and illuminated conditions. The six pH
conditions investigated fall within the range of pH values for cloud water previously measured
at Tai Mo Shan (pH 3.0 to 5.9) (Li et al., 2020). The pH of the artificial cloud water used to
suspend the bacterial cells was adjusted using sodium hydroxide and hydrochloric acid. Table
S1 shows the resulting concentrations of organic and inorganic ions in the artificial cloud water
used in these experiments, which are similar to those in cloud water collected at Tai Mo Shan
by Li et al. (2020).
During some experiments, aliquots of the solutions (10 mL) were taken at time points
0 h, 2 h, 4 h, 8 h, and 12 h and analyzed by ultra-performance liquid chromatography-mass
spectrometry (UPLC-MS). Each aliquot of solution was first passed through a 0.22 µm filter
to remove intact bacterial cells. Water-insoluble and water-soluble biological material and
organic compounds were then extracted from these filtered solutions using the method
described in Section S2. 200 µL of the extract was then transferred into glass vial inserts for
UPLC-MS analysis. Non-targeted UPLC-MS analysis was performed using an ultrahigh
performance liquid chromatography system (ExionLC AD system, Sciex) coupled to a high-
resolution quadrupole-time-of-flight mass spectrometer (TripleTOF 6600 system, Sciex)
equipped with electrospray ionization (ESI). Chromatographic separation was performed on a
Kinetex HILIC LC column (100 × 2.1 mm, 2.6 µm, 100 Å, Phenomenex) using positive ESI
mode. Since very low signals were obtained for negative ESI mode, we did not use it for our
analysis. Details about the UPLC-MS operation, data processing, and statistical analysis can
be found in Section S3.
**2.4. Investigations of the biodegradation of organic acids at different pH under**
**illuminated vs. dark conditions**
The biodegradation of seven organic acids (formic acid, acetic acid, oxalic acid, maleaic
acid, malonic acid, glutaric acid, and methanesulfonic acid (MSA)) that were mixed together
were measured at pH 4.3 and pH 5.9 under both dark and illuminated conditions. The
concentrations for each of the forementioned organic acids in cloud water and rain water
typically fall within the range of 1 to 10 µM (Tsai and Kuo, 2013; Löflund et al., 2002; Sun et
al., 2016; Li et al., 2020). Due to the detection limits of the IC system used to measure the
organic acids, the concentration for each organic acid was set to 50 µM (Table S2), which is
around 10 times higher than the concentrations typically measured in cloud water. The
concentrations of inorganic ions in the artificial cloud water were also increased by 10 times.
Vaitilingom et al. (2010) previously reported that the same biodegradation rates will be
obtained as long as the concentration ratio of the chemical compounds to bacterial cells is
constant. However, the authors drew this conclusion based on experiments performed using a
*Pseudomonas graminis* bacterial strain incubated in the presence of a single organic compound
as the carbon source. At present, it is unclear whether this conclusion can be extrapolated to
other bacteria species incubated in the presence of multiple organic compounds, and this
warrants further study. Nevertheless, we made the same assumption (i.e., the same
biodegradation rates will be obtained as long as the concentration ratio of the chemical
compounds to bacterial cells is constant) as was done in previous studies that investigated the
biodegradation of multiple organic compounds by different bacteria species (Vaïtilingom et al.,
2011; Jaber et al., 2020; Jaber et al., 2021). Hence, the bacteria concentration used was set to
$10^6$ cells mL$^{-1}$ to maintain the same concentration ratio of the organic acids to bacterial cells.
Table S2 shows the resulting concentrations of the organic and inorganic ions in the artificial
cloud water used in these experiments.

During each experiment, aliquots of the solutions (0.6 mL) were taken every 2 hours

over 12 hours. The organic acid concentrations in each filtered aliquot of solution were
measured by ion chromatography (IC) using a Dionex ICS-1100 (ThermoFisher Scientific)
system. Details of the IC operation can be found in Section S4. To calculate the initial
biodegradation rate, the time evolution of each organic acid concentration over 12 h was plotted
and fitted with the following equation (Vaïtilingom et al., 2011; Jaber et al., 2020; Jaber et al.,

2021):

$$\ln \left( \frac{C}{C_0} \right) = f(t) = -k \times t \tag{1}$$

where $k$ $(s^{-1})$ is the rate constant obtained from the exponential fit to the decay of the organic
acid. The following equation was used to calculate the biodegradation rate per bacteria cell (R):

$$R = \frac{k \times C_0}{[Cell]_{experiment}}, (mol\ cell^{-1}s^{-1}) \tag{2}$$

where $C_0$ $(mol \cdot L^{-1})$ is the initial concentration of the organic acid, $[Cell]_{experiment}$ $(cell \cdot$
$L^{-1})$ is the concentration of bacterial cells in the experiment. Control experiments were
performed under illuminated and dark conditions using solutions that contained organic acids
but no bacterial cells. The organic acids did not degrade in these control experiments.
**3. Results and discussion**
**3.1. Impact of pH on the survival and energetic metabolism of bacteria under illuminated**
**and dark conditions**

Figure 1 shows the survival rates and ADP/ATP ratios of the *E. hormaechei* B0910 and

*E. hormaechei* pf0910 strains over time under illuminated and dark conditions at different
artificial cloud water pH. The ADP/ATP ratio is used as an indicator of the bacteria's metabolic
activity and survival rate in this study. Growing cells usually maintain a constant ADP/ATP
ratio because whenever there is a decrease in intracellular ATP production, its degradation
product ADP will be resynthesized to form ATP to maintain intracellular ATP concentrations
(Koutny et al., 2006; Guan and Liu, 2020). In contrast, when there is a disruption in the
metabolism of ATP production, ATP cannot be resynthesized from ADP even though ATP is
still converted to ADP, which will cause the ADP/ATP ratio to increase (Koutny et al., 2006;
Guan and Liu, 2020).
The artificial cloud water pH clearly had a significant effect on the survival rates and
ADP/ATP ratios of the two strains. At pH 3.3, the concentrations of viable cells decreased to
zero after 20 minutes regardless whether the strains were exposed to light. For pH 4.3, 4.5 and
4.7, the survival and ADP/ATP ratios of the two strains depended on whether they were
exposed to light. There were no significant changes in the survival rates and ADP/ATP ratios
for both strains under dark conditions. In contrast, the concentrations of viable cells for both
strains gradually decreased when they were exposed to light. Consistent with the lower survival
rates, the ADP/ATP ratios for both strains increased over time. The survival rates and
ADP/ATP ratios were the lowest and highest, respectively, at pH 4.3 after 12 h of illumination.
There were no significant changes in the survival rates and ADP/ATP ratios of both strains at
pH 5.2 and 5.9 under illuminated and dark conditions.

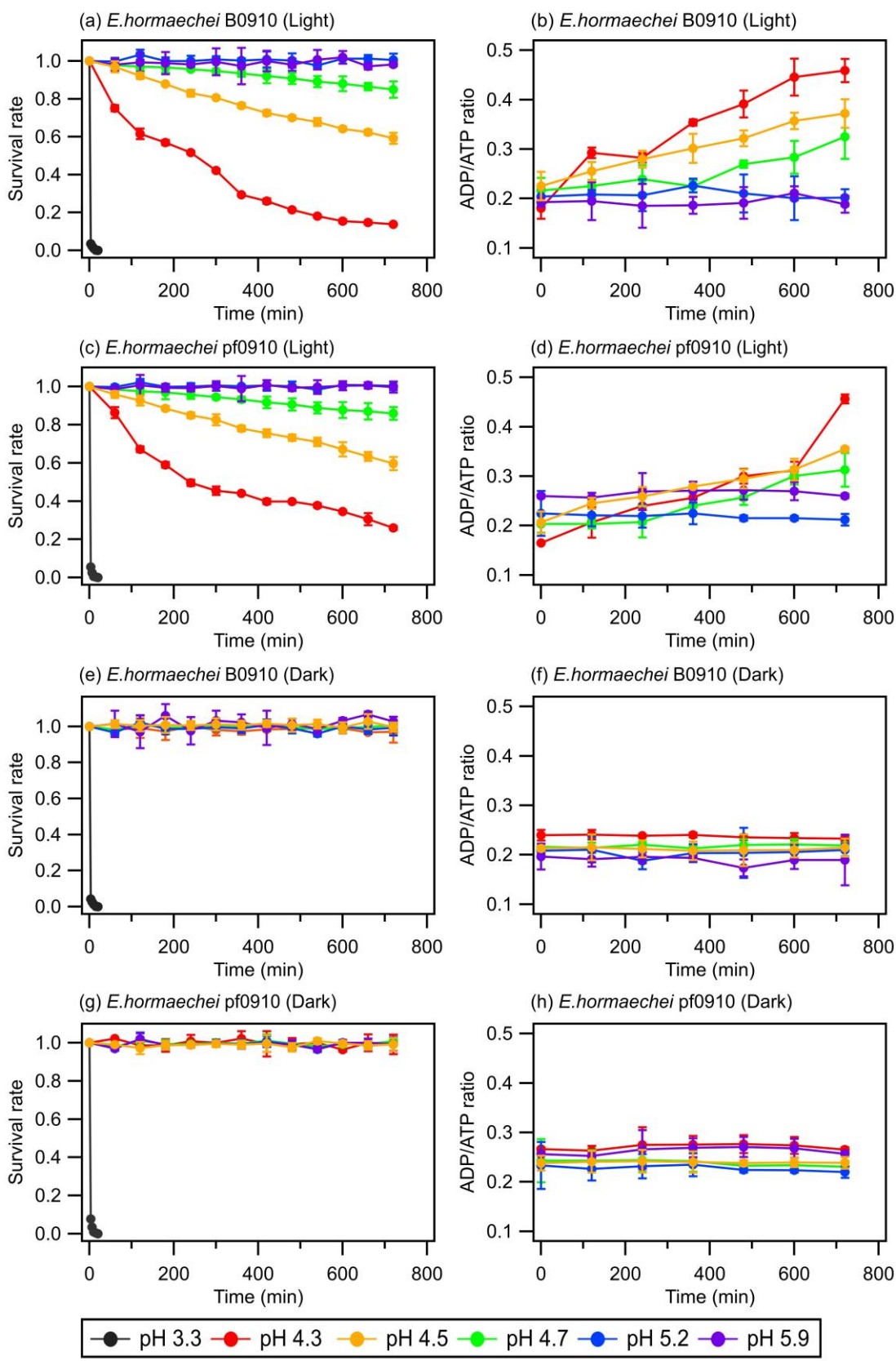


**Figure 1.** Survival rates and ADP/ATP ratios of the *E. hormaechei* B0910 and *E. hormaechei* pf0910 strains at pH 3.3 to pH 5.9 under illuminated and dark conditions over time. The

survival rate is defined as the number concentration of culturable viable cells divided by the
initial number concentration of culturable viable cells at time point 0 min. Error bars represent
one standard deviation from the mean of biological triplicates.

Figure 1 clearly shows that the artificial cloud water pH and exposure to light can have

a synergistic effect on the survival and energetic metabolism of *E. hormaechei* B0910 and *E.*
*hormaechei* pf0910. Based on these results, both strains will likely survive during the daytime
and nighttime in pH > 5 cloud water. However, cloud water pH will play an important role in
dictating the fraction of the bacteria that will survive in the daytime at pH 4 to 5. A low pH
environment can lower the internal pH of cells, which affects essential pH-dependent biological
and cellular functions such as decreased enzymatic activity, compromised cellular processes
(e.g., central metabolic pathways, ATP production), and protein denaturation in cells (Bearson
et al., 1997; Lund et al., 2014). Our genomic analysis revealed that the two strains have genes
encoding a F1F0-type ATP synthase, which can export protons from their cytoplasm to cope
with pH stress (Krulwich et al., 2011). In addition, genes encoding potassium transporters,
which may be involved in pH homeostasis (i.e., both Kup-type low-affinity and Kdp-type high-
affinity potassium transporters) (Brzoska et al., 2022) were found in the genome of both strains
(Table S3). Our results indicated that both strains will likely survive in pH 4 to 5 cloud water
at night. However, being in cloud water at pH 4 to 5 will likely negatively impact the ability of
cells to tolerate sunlight, which will affect their survival during the daytime. Based on our
results, we estimate that the half-lives of the bacteria strains in pH 4.3 cloud water under
illumination conditions (e.g., light intensity, wavelengths) similar to those in our study are
around 430 min. The half-lives of the bacteria strains in pH < 4 are cloud water are lower.
Based on our results, we estimate that the daytime and nighttime half-lives of the bacteria
strains in pH 3.3 cloud water are around 2 min.
**3.2. Compounds released by bacteria under acidic and illuminated conditions**

Some bacteria species adapt to sunlight exposure and acidic environments by deploying

adaptation strategies and defensive mechanisms such as undergoing DNA repair, aggregation-
promoting, and pigmentation mechanisms (Bearson et al., 1997; Davey and O'toole, 2000;
Delort et al., 2010; Flemming and Wingender, 2010; Vaïtilingom et al., 2012; Matulova et al.,
2014; Guan and Liu, 2020). Some of these adaptation strategies and defensive mechanisms will
cause the bacteria to release organic compounds into cloud water (Davey and O'toole, 2000;
Delort et al., 2010; Flemming and Wingender, 2010; Vaïtilingom et al., 2012; Matulova et al.,
2014). In addition, bacterial cellular damage and lysis will lead to the release of biological
material and organic compounds. To investigate the compounds released by *E. hormaechei*
B0910 and *E. hormaechei* pf0910 during exposure to light and acidic environments, we used
UPLC-MS to analyze the solutions in experiments where pH 4.3 and pH 5.9 artificial cloud
water were used. The UPLC-MS measurements revealed that cell lysis led to the release of
water-soluble and water-insoluble compounds when the two strains were exposed to light at
pH 4.3. The quantities of these compounds changed with light exposure time. In contrast, no
water-soluble and water-insoluble compounds were detected in the solutions of the two strains
under dark conditions at pH 4.3, and under dark and illuminated conditions at pH 5.9. This
suggested that these two strains did not release organic compounds and the cells remained
intact under these conditions. It is also possible that these two strains released organic
compounds as an adaption strategy and/or defensive mechanism but the concentrations of these
compounds were below the detection limits of our UPLC-MS instrument.
Principal component analysis (PCA) with 95% confidence ellipse was applied to the
UPLC-MS data of the detected water-soluble and water-insoluble compounds to identify
discriminations between samples with different light exposure times. In each PCA plot (Figure
2), samples with the same light exposure time clustered together. While there was slight overlap
between some of the clusters in the PCA plots, the clusters were mostly separated from one
another. Partial least squares discrimination analysis (PLS-DA) was applied to the UPLC-MS
data to identify water-soluble and water-insoluble compounds that showed significant changes
in their relative abundances during exposure to light. 259 water-soluble compounds and 215
water-insoluble compounds were identified for *E. hormaechei* B0910 (Figure S3), while 209
water-soluble compounds and 251 water-insoluble compounds were identified for *E.*
*hormaechei* pf0910 (Figure S4). We identified the molecular formulas and chemical structures
of 78 water-soluble compounds and 144 water-insoluble compounds released by *E. hormaechei*
B0910, and 118 water-soluble compounds and 114 water-insoluble compounds released by *E.*
*hormaechei* pf0910. These identified compounds were subsequently classified into different
classes based on their chemical functionalities.

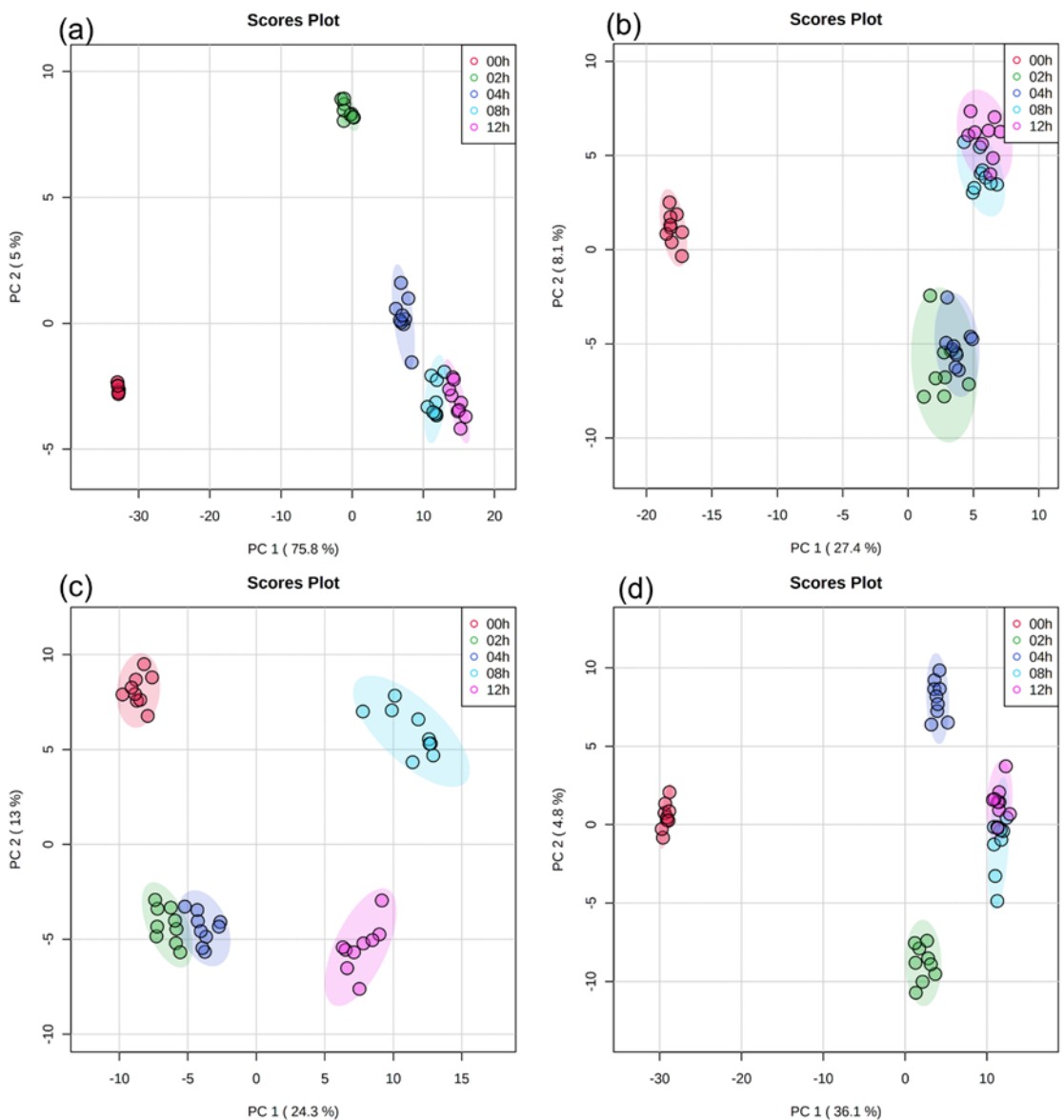


**Figure 2.** PCA results of UPLC-MS data: (a) water-soluble compounds and (b) water-insoluble

compounds from *E. hormaechei* B0910, and (c) water-soluble compounds and (d) water-

insoluble compounds from *E. hormaechei* pf0910 during exposure to light at pH 4.3. Each

cluster representing a different light exposure time (i.e., 0 h, 2 h, 4 h, 8 h, and 12 h) has nine

points since three samples were taken at each light exposure time, and UPLC-MS analysis was

performed in triplicate for each sample.

Figures 3 and S5 show the time evolution of the UPLC-MS total ion chromatograph
(TIC) signals of the different classes of water-soluble and water-insoluble compounds released
by *E. hormaechei* B0910 and *E. hormaechei* pf0910 over time, respectively. The UPLC-MS
TIC signals of the classes of water-soluble and water-insoluble compounds released by the two
strains increased with light exposure time. The increase in the UPLC-MS TIC signals coincided
with the decrease in the bacteria survival rate and the increase in the ADP/ATP ratio. Even
though the heatmaps showed that some of the compounds had noticeable changes in their
relative abundances during exposure to light (Figures S3 and S4), the relative abundances of
the different classes of compounds contributed to the total TIC at each time point did not change
substantially (Figures S6 and S7).

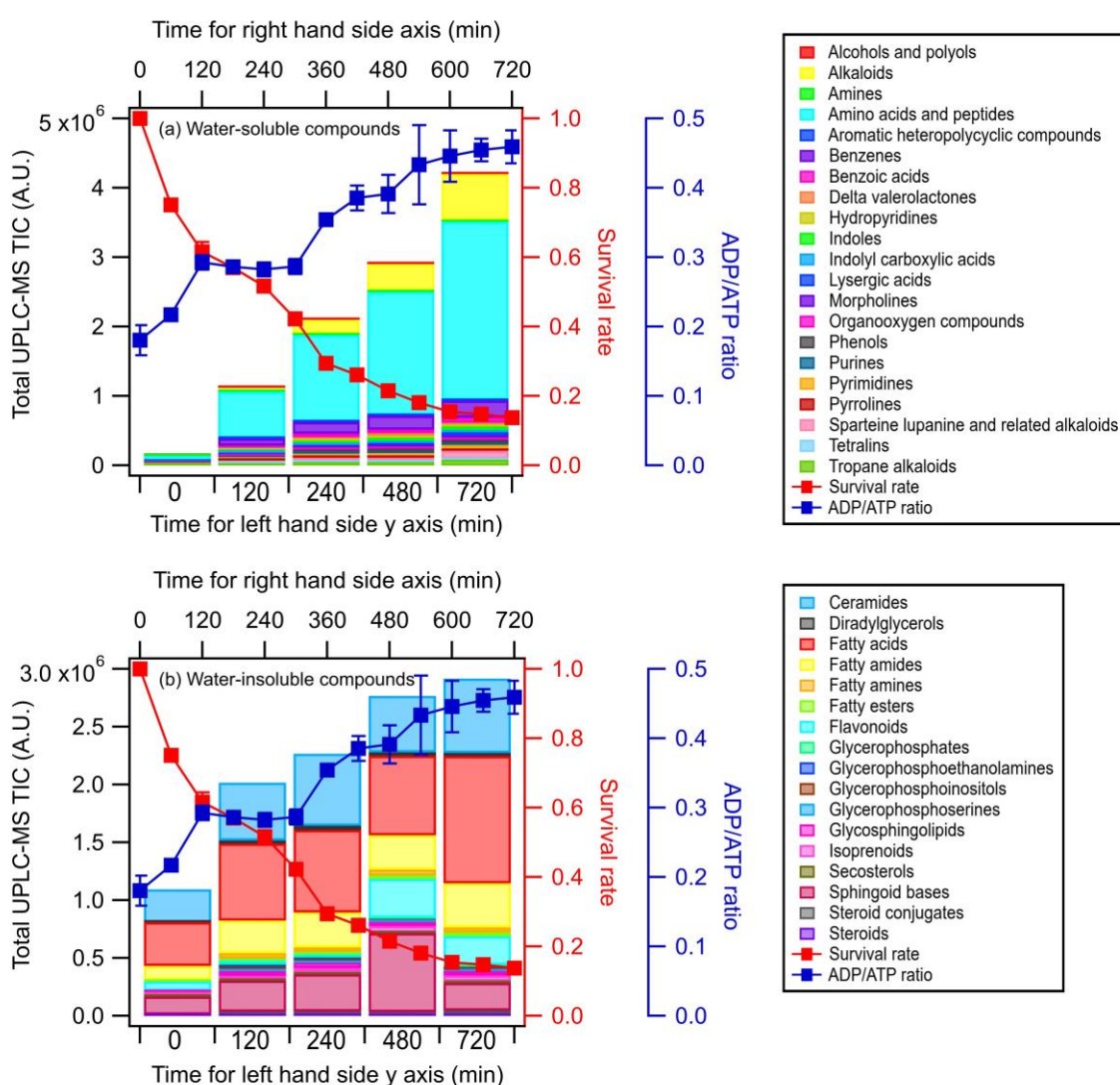


**Figure 3.** Time evolution of the UPLC-MS total ion chromatograph (TIC) signals of (a) water-soluble compounds, and (b) water-insoluble compounds from *E. hormaechei* B0910 during exposure to light at pH 4.3 over time. These compounds are classified based on their chemical functionality. Also shown are the time evolution of the survival rate and ADP/ATP ratio of *E. hormaechei* B0910.

To better understand the compounds released by the two strains, the O/C and H/C elemental ratios of the identified compounds were used to construct Van Krevelen (VK) diagrams. Regions of the VK diagrams were assigned to eight chemical classes based on the combined O/C and H/C ratios: lipids, unsaturated hydrocarbons, condensed aromatic structures, peptides, lignin, tannin, amino sugars, and carbohydrates (Table S4) (Bianco et al., 2018; Laszakovits and Mackay, 2022). Rivas-Ubach et al. (2018) previously reported that the region of the VK diagram assigned to amino sugars overlaps with the region for nucleic acids. Figures S8 and S9 show the VK diagrams for water-soluble and water-insoluble compounds released by *E. hormaechei* B0910, respectively, while Figures S10 and S11 show the VK diagrams for water-soluble and water-insoluble compounds released by *E. hormaechei* pf0910, respectively. Majority of the water-soluble and water-insoluble compounds released from both strains (50% to 60%) were assigned as lipids based on their O/C and H/C ratios, while the second most abundant compound class was peptides (10% to 20%). The two least abundant compound classes were amino sugars/nucleic acids and carbohydrates. Since the dry matter of a typical bacterial cell contains approximately 55% proteins and amino acids, 24% nucleic acids, 10% carbohydrates, 7% lipids, and 5% inorganic minerals and trace elements (Watson et al., 2007), the differences in the abundance of compound classes detected vs. the dry matter of a typical bacterial cell indicated that cellular components were likely biologically and/or chemically modified during and after cell lysis during exposure to light. For instance, the large abundance of peptides detected could be a result of biological and/or chemical modifications of proteins and amino acids, which comprise majority of the dry matter of a typical bacterial cell. Peptide bonds are formed by biochemical reactions where a water molecule is removed as the amino group of one amino acid is joined to the carboxyl group of a neighboring amino acid. The large abundance of lipids was unsurprising since lipids are the main component of cell

membranes so large quantities of lipids are expected from the lysed cells. Most of the lipid molecules released during cell lysis may not have undergone biological and/or chemical modifications under our experimental conditions. The two least abundant compound classes were amino sugars/nucleic acids and carbohydrates. This was somewhat surprising since nucleic acids and carbohydrates are abundant in the dry matter of a typical bacterial cell. It is possible that these compounds were biologically and/or chemically modified to form other compounds (e.g., exopolymeric substances) during exposure to light (Matulova et al., 2014). In addition, the extraction procedure employed (Section S2) may not have extracted these compounds effectively for analysis. For instance, nucleic acids and carbohydrates are polar molecules, which are difficult to retain on the solid phase extraction columns used in this study. These compounds may also have been poorly separated in UPLC and/or inefficiently ionized by ESI.

These detected compounds indicated that bacterial cell lysis could be a source for carbon in cloud water. Many of the compound classes detected in this study have previously been measured in atmospheric cloud water. For instance, large abundances of peptide-like compounds and lipid-like compounds have been measured in cloud water from Puy de Dôme (Bianco et al., 2018; Bianco et al., 2019), which is consistent with the detection of large abundances of compounds assigned to the peptide and lipid compound classes in this study. This suggested that peptide-like and lipid-like compounds could be used as biomarkers to evaluate bacterial contributions to atmospheric samples. Previous studies have used fatty acids, which are integral building blocks of lipids, in atmospheric samples as biomarkers for characterizing and quantifying bacteria, and assessing the atmospheric transport of bacteria (Kawamura et al., 2003; Lee et al., 2004; Tyagi et al., 2015). While this study shows that bacterial cell lysis will release large quantities of peptide-like and lipid-like compounds, using these compounds as biomarkers for bacterial cell lysis in atmospheric samples will likely be complex as the concentrations of these compounds will likely change with time. This is because peptide-like and lipid-like compounds will undergo chemical and biological transformations after they have been released during cell lysis, which will impact their concentrations in atmospheric samples. Amino acids, which are building blocks of peptides, are known to

undergo chemical reactions with oxidants in cloud water, (Bianco et al., 2016). In addition, peptide-like and lipid-like compounds can be produced and/or consumed by cloud microorganisms to maintain their metabolism (Bianco et al., 2019; Jaber et al., 2021).

**3.3. Impact of pH on the biodegradation of organic acids by bacteria under illuminated and dark conditions**

The biodegradation of seven organic acids (i.e., formic acid, acetic acid, oxalic acid, maleic acid, malonic acid, glutaric acid and MSA) that were mixed together were measured under dark and illuminated conditions at pH 4.3 and pH 5.9. Only some of the seven organic acids were biodegraded by the two strains. Based on our experimental conditions (liquid water content $\approx 10^{12}$ µg m$^{-3}$, the density of water) and the organic acids' Henry's law constants, these organic acids will be in the aqueous phase and are not expected to volatilize during these experiments. Thus, the observed decays were due to bacterial metabolism. *E. hormaechei* B0910 biodegraded formate and oxalate under dark and illuminated conditions at pH 4.3 and pH 5.9, and biodegraded malonate and maleate only under dark conditions at pH 4.3 and pH 5.9. In contrast, *E. hormaechei* pf0910 biodegraded only formate and oxalate under dark and illuminated conditions at pH 4.3 and pH 5.9. Biodegradation was not observed for acetate, MSA, and glutarate.

Table S5 summarizes the enzymes or metabolic pathways related to the biodegradation of organic acids in the two strains. Genes encoding formate dehydrogenases were identified in both genomes, which is consistent with the observed formate biodegradation. However, no known genes for oxalic acid biodegradation (Liu et al., 2021) were found in the genomes of both strains, which suggested the presence of yet to be characterized pathways that catalyzed the biodegradation. Interestingly, a protein with Cupin 2 domain was found in both genomes. The Cupin superfamily consists of a diverse range of enzymes including oxalate oxidase and oxalate decarboxylase that can biodegrade oxalic acid (Burrell et al., 2007).

Only the *E. hormaechei* B0910 strain was observed to biodegrade malonic acid. Interestingly, the malonyl-CoA-acyl carrier transcacylase observed in the *E. hormaechei* pf0910 strain seems to be a fusion protein, which may render it ineffective in utilizing malonic

acid. Although no gene encoding maleate isomerase was identified in the genomes of both
strains, the maleic acid biodegradation observed can be attributed to the activity of other
enzymes with broad substrates specificity (Hatakeyama et al., 2000). The genes encoding for
the small and large protein subunits that together form the 3-isopropylmalate dehydratase, the
enzyme that isomerizes 2-isopropylmalate to 3-isopropylmalate, were found in both the
*Enterobacter* strains. The small and large protein subunits of this enzyme are homologous to
the small (51% amino acid identity) and large (59% amino acid identity) protein subunit
constituents of maleate hydratase (HbzIJ) from *Pseudomonas alcaligenes* NCIMB 9867 that
converts maleate to D-malate (Liu et al., 2015). Given the high protein homology, we speculate
that the 3-isopropylmalate dehydratase in the *Enterobacter* strains may have a broader substrate
specificity than known and it may be able to biodegrade maleate.

The lack of biodegradation of acetic acid, MSA, and glutaric acid in the experiments

could be partly explained by the genomic information. Both strains have genes that encode
enzymes involved in the biodegradation (Table S5) and associated uptake transporters (i.e.,
acetate permease (ActP) and succinate-acetate/proton symporter (SatP)) of acetic acid. The
lack of the corresponding biodegradation in the experiments could be due to the low uptake of
acetic acid by cells as ActP functions to scavenge low concentrations of the compound
(Gimenez et al., 2003) while SatP could be inhibited by formic acid found in the cloud water
medium (Sá-Pessoa et al., 2013). Genes encoding the two-component alkanesulfonate
monooxygenase for MSA biodegradation were found in both strains, but they were likely not
expressed as sulfur was not deficient in the cloud water medium (Kahnert et al., 2000; Eichhorn
and Leisinger, 2001), which is consistent with the absence of MSA biodegradation in the
experiments. While genes encoding succinate-semialdehyde dehydrogenase/glutarate-
semialdehyde dehydrogenase, which display a reversible conversion between glutarate-
semialdehyde and glutarate in the KEGG database (Kanehisa et al., 2022), were found in both
strains, to the best of our knowledge there is no report of experimental results confirming that
the reaction can go in the reverse direction from glutarate to glutarate-semialdehyde. In
addition, a study of glutaric semialdehyde dehydrogenase reported the irreversible nature of
the catalysis of glutarate semialdehyde to glutarate (Ichihara and Ichihara, 1961). Thus, it is
not surprising that glutarate biodegradation was not observed for the two strains.

Figure 4 summarizes the measured biodegradation rates of the organic acids for the two

strains under dark and illuminated conditions at pH 4.3 and pH 5.9. These biodegradation rates
were determined from fits to the decays of the organic acids from reaction time 0 to 12 hour in
each experiment (Section 2.4). The measured biodegradation rates were around $10^{-19}$ to $10^{-18}$
mol cell$^{-1}$ s$^{-1}$, which were on the same order of magnitude as the bacterial strains isolated from
cloud water and implemented into cloud models (Vaitilingom et al., 2010; Vaïtilingom et al.,
2011; Fankhauser et al., 2019). Although both strains were affiliated to *E. hormaechei*, the
artificial cloud water pH and exposure to light impacted their biodegradation of organic acids
differently. The rates at which formate and oxalate were biodegraded by *E. hormaechei* B0910
had the following order: dark conditions at pH 5.9 > illuminated conditions at pH 5.9 > dark
conditions at pH 4.3 > illuminated conditions at pH 4.3. This order was different for *E.*
*hormaechei* pf0910: dark conditions at pH 5.9 > dark conditions at pH 4.3 > illuminated
conditions at pH 5.9 > illuminated conditions at pH 4.3. Despite the effects that the artificial
cloud water pH and exposure to light had on the formate and oxalate biodegradation, the fastest
and slowest biodegradation rates only differed by a factor of 1.4 to 3.7. Figure S12 compares
the biodegradation rates measured at pH 4.3 vs. pH 5.9, and under illuminated vs. dark
conditions. For the effect of artificial cloud water pH on the biodegradation of organic acids by
*E. hormaechei* B0910, the differences in the biodegradation rates were statistically significant
for the four acids (Student's t test, $p$ value < 0.05). Conversely, the differences in the
biodegradation rates of formate and oxalate as a result of light exposure were statistically
significant at pH 5.9 (Student's t test, $p$ value < 0.05). For the effect of artificial cloud water
pH on the biodegradation of organic acids by *E. hormaechei* pf0910, only the difference in the
dark biodegradation of oxalate was statistically significant (Student's t test, $p$ value < 0.05). In
contrast, light exposure reduced the formate biodegradation rates significantly at both pH 4.3
and pH 5.9 (Student's t test, $p$ value < 0.05), and the oxalate biodegradation rate significantly
at pH 5.9 (Student's t test, $p$ value < 0.05).

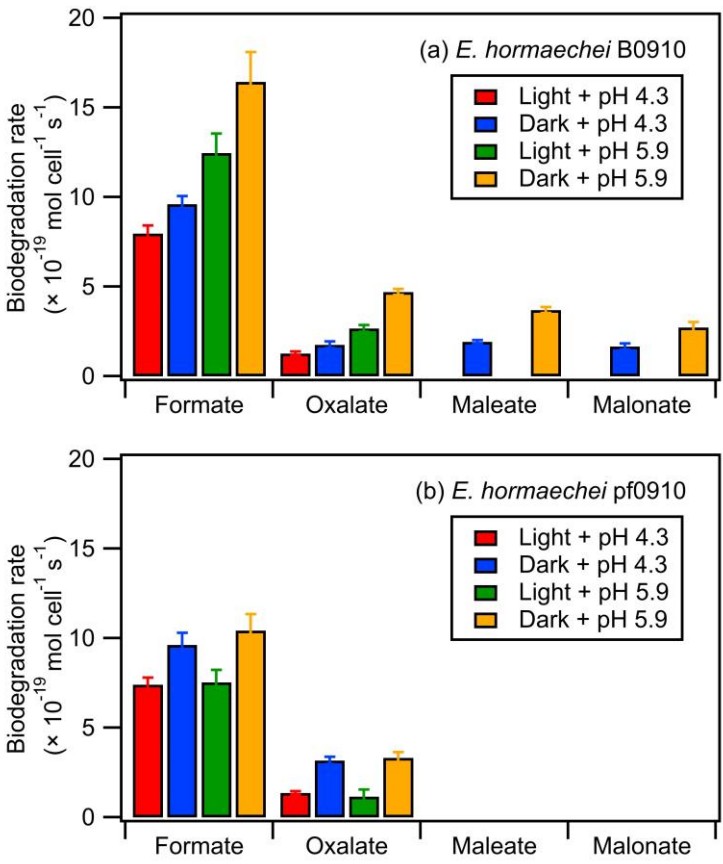


**Figure 4.** Biodegradation rates of oxalate, maleate, and malonate by (a) *E. hormaechei* B0910

and (b) *E. hormaechei* pf0910 under light and dark conditions at pH 4.3 and pH 5.9. Error bars

represent one standard deviation from the mean biodegradation rate.

The survival rates and ADP/ATP ratios of both strains were also monitored during the

biodegradation experiments (Figure S13). There were no significant changes in the survival

rates and ADP/ATP ratios of both strains during the biodegradation process under dark

conditions at pH 4.3, as well as under dark and illuminated conditions at pH 5.9. In contrast,

the concentrations of viable cells gradually decreased until only 48% and 60% of the initial

concentrations of viable cells remained at 12 h for *E. hormaechei* B0910 and *E. hormaechei*

pf0910, respectively, during exposure to light at pH 4.3. The ADP/ATP ratios for both strains

also increased during this time period, consistent with the lower metabolic activity and lower

survival rate.

A simple kinetic analysis was performed to identify the factors that will impact the relative contributions of bacterial activity vs. $\cdot$OH/NO$_3\cdot$ chemistry in cloud water during the daytime and nighttime. Details of the calculations performed in this kinetic analysis can be found in Section S5. Our approach of considering daytime and nighttime processes separately was different from the approach used by previous studies, which determined the relative contributions of bacterial activity and chemical reactions on the degradation of organic compounds by only considering dark biodegradation processes and $\cdot$OH photochemical reactions (Vaïtilingom et al., 2011; Jaber et al., 2020; Jaber et al., 2021). Here, biodegradation rates that were measured under illuminated conditions were used for the daytime scenario, while biodegradation rates that were measured under dark conditions were used for the nighttime scenario. We used the average of biodegradation rates measured for the two strains for our calculations. Formate, oxalate, and malonate were chosen for our analysis since their $\cdot$OH and NO$_3\cdot$ reaction rate constants were available in the literature. $\cdot$OH and NO$_3\cdot$ are the main tropospheric aqueous-phase free radicals during the daytime and nighttime, respectively (Herrmann et al., 2010). The average measured biodegradation rates of formate, oxalate, and malonate were first converted to biodegradation rate constants. These biodegradation rate constants and the corresponding $\cdot$OH and NO$_3\cdot$ reaction rate constants provided by the literature (Table 1) were subsequently used for calculations of the biodegradation rates and chemical reaction rates in cloud water (Section S5). A bacteria concentration of $8 \times 10^7$ cell L$^{-1}$ was assumed in our calculations for the daytime scenario at pH ~5 and the nighttime scenarios at pH ~4 and ~5, which was the same bacteria concentration used in previous studies and represented the highest estimate of actual live bacteria concentrations (i.e., 100% of metabolically active cells) (Vaïtilingom et al., 2011; Jaber et al., 2020; Jaber et al., 2021). Based on our investigations of the survival and energetic metabolism of bacteria under illuminated conditions at pH 4 to 5 (Figure 1), we expect the bacteria concentrations to gradually decrease for the daytime scenario at pH ~4. Thus, for simplicity, we assumed a lower bacteria concentration in our calculations for the daytime scenario at pH ~4, whereby we multiplied the bacteria concentration of $8 \times 10^7$ cell L$^{-1}$ by a factor of 0.75. This factor was obtained by taking the average survival rates for the two strains from reaction time 0 to 12 hour in our experiments conducted under illuminated conditions at pH 4.3 (Figure S13). The rates of oxidation by $\cdot$OH

and $NO_3 \cdot$ chemical reactions will depend on their respective concentrations. Hence, we used
the average $\cdot OH$ and $NO_3 \cdot$ concentrations reported by Herrmann et al. (2010) for remote,
marine, and urban environments in our calculations (Table S6) (Herrmann et al., 2010).
**Table 1.** Rate constants used to estimate the loss rates by biodegradation and chemical reactions
(i.e., $\cdot OH$ oxidation (daytime) and $NO_3 \cdot$ (nighttime)).

| Rate constant (Daytime) | | | | |
|---|---|---|---|---|
| Reaction | | Formic | Oxalic | Reference |
| Chemical | $k_{OH,Acid}$ $(L\ mol^{-1}\ s^{-1})$ | $2.40 \times 10^9$ | $1.60 \times 10^8$ | (Ervens et al., 2003) |
| Biodegradation | $k_{cell,acid}$ (pH ~4) $(L\ cell^{-1} s^{-1})$ | $1.53 \times 10^{-13}$ | $2.65 \times 10^{-15}$ | This study |
| | $k_{cell,acid}$ (pH ~5) $(L\ cell^{-1} s^{-1})$ | $1.92 \times 10^{-13}$ | $2.36 \times 10^{-14}$ | This study |


| Rate constant (Nighttime) | | | | | |
|---|---|---|---|---|---|
| Reaction | | Formate | Oxalate | Malonate | Reference |
| Chemical | $k_{NO_3,Acid}$ $(L\ mol^{-1}\ s^{-1})$ | $4.20 \times 10^7$ | $4.40 \times 10^7$ | $5.60 \times 10^6$ | (Herrmann et al., 2010) |
| Biodegradation | $k_{cell,acid}$ (pH ~4) $(L\ cell^{-1} s^{-1})$ | $1.92 \times 10^{-13}$ | $5.18 \times 10^{-15}$ | $2.81 \times 10^{-15}$ | This study |
| | $k_{cell,acid}$ (pH ~5) $(L\ cell^{-1} s^{-1})$ | $2.59 \times 10^{-13}$ | $7.80 \times 10^{-14}$ | $4.55 \times 10^{-14}$ | This study |

Calculations were performed for a variety of remote, marine, and urban environments
with different formate, oxalate, and malonate concentrations that were previously reported in
the literature (Table S7). Figure 5 shows the predicted relative contributions of bacterial
activity vs. $\cdot OH/NO_3 \cdot$ chemistry in remote, marine, and urban environments. $\cdot OH$
photochemistry will make a larger contribution to the daytime degradation of formate and
oxalate in remote and marine environments due to the high $\cdot OH$ concentrations in these
environments ($2.2 \times 10^{-14}$ M and $2 \times 10^{-12}$ M, respectively). In contrast, bacterial activity will
play a bigger role in the daytime degradation of formate in urban environments due to their
lower $\cdot OH$ concentrations ($3.5 \times 10^{-15}$ M). However, $\cdot OH$ photochemistry will play a larger
role in the daytime degradation of oxalate in urban environments due to the slow oxalate
biodegradation rates. The low nighttime $NO_3 \cdot$ concentrations in remote and marine
environments ($5.1 \times 10^{-15}$ M and $6.9 \times 10^{-15}$ M, respectively) will result in bacterial activity
playing a bigger role in the nighttime degradation of formate, oxalate, and malonate in these
two environments. In urban environments, bacterial activity will play a bigger role in the
nighttime degradation of formate, but the nighttime degradation of oxalate and malonate will
be dominated by $NO_3 \cdot$ chemistry due to the slow biodegradation rates of oxalate and malonate.

Our simple kinetic analysis indicated that the organic acid, cloud water pH, radical

oxidant concentration, and time of day (i.e., daytime vs. nighttime) will impact the relative
contributions of bacterial activity vs. $\cdot OH/NO_3 \cdot$ chemistry in the aqueous phase. However,
there are a number of caveats that should be noted. First, the biodegradation rates used in this
analysis were from experiments conducted at 25 °C, which may be more representative of
warmer regions during the summer (e.g., Hong Kong and parts of South China). Slower
biodegradation rates will likely be measured at lower temperatures (Ariya et al., 2002;
Vaitilingom et al., 2010; Husárová et al., 2011; Vaïtilingom et al., 2011), which will impact
the relative contributions of bacterial activity vs. $\cdot OH/NO_3 \cdot$ chemistry. Second, our analysis
did not account for how the presence of aqueous-phase oxidants (e.g., $\cdot OH$ in the daytime,
$NO_3 \cdot$ in the nighttime) will impact the survival and energetic metabolism of bacteria, which in
turn will impact the relative contributions of bacterial activity vs. $\cdot OH/NO_3 \cdot$ chemistry. Third,
our analysis did not account for the physical separation of cloud droplets containing bacteria
cells from cell-free cloud droplets. Only a small fraction of cloud droplets will contain
metabolically active bacteria cells, and the bacterial metabolism cannot affect the composition
of organic acids in cell-free cloud droplets (Fankhauser et al., 2019; Khaled et al., 2021).
Hence, only $\cdot OH/NO_3 \cdot$ chemistry will govern the degradation of organic acids in cell-free
droplets. Consequently, not accounting for the physical separation of cloud droplets containing
bacteria cells from cell-free cloud droplets will result in an overestimation of the overall
contribution of bacterial activity to the biodegradation of organic compounds (Fankhauser et
al., 2019; Khaled et al., 2021). Fourth, our analysis only considers biodegradation and chemical
reactions occurring in the aqueous phase and ignores gas-aqueous phase exchanges and gas-
phase chemical reactions. Nah et al. (2018) previously showed that the gas-aqueous phase
partitioning of organic acids will depend on the organic acid's Henry's law constant and acid

dissociation constants, liquid water concentration, temperature, and pH (Section S6). Figure S14 shows that a significant fraction of formic acid will be in the gas phase at pH 4 and 5 under cloud water conditions, whereas all of oxalic acid, malonic acid, and maleic acid will be in the aqueous phase at pH 4 and 5 under cloud water conditions. This suggests that gas-phase chemical reactions will likely play an important role in consuming formic acid, whereas the consumption of oxalic acid, malonic acid, and maleic acid will likely mainly be through bacterial activity and chemical reactions in the aqueous phase. Quantifying the exact contributions of aqueous-phase bacterial activity vs. aqueous-phase $\cdot OH/NO_3\cdot$ chemistry vs. gas-phase $\cdot OH/NO_3\cdot$ chemistry under different cloud water pH conditions will require a multi-phase box model similar to the one used by Khaled et al. (2021). This is beyond the scope of the current study but can be a subject of future studies.

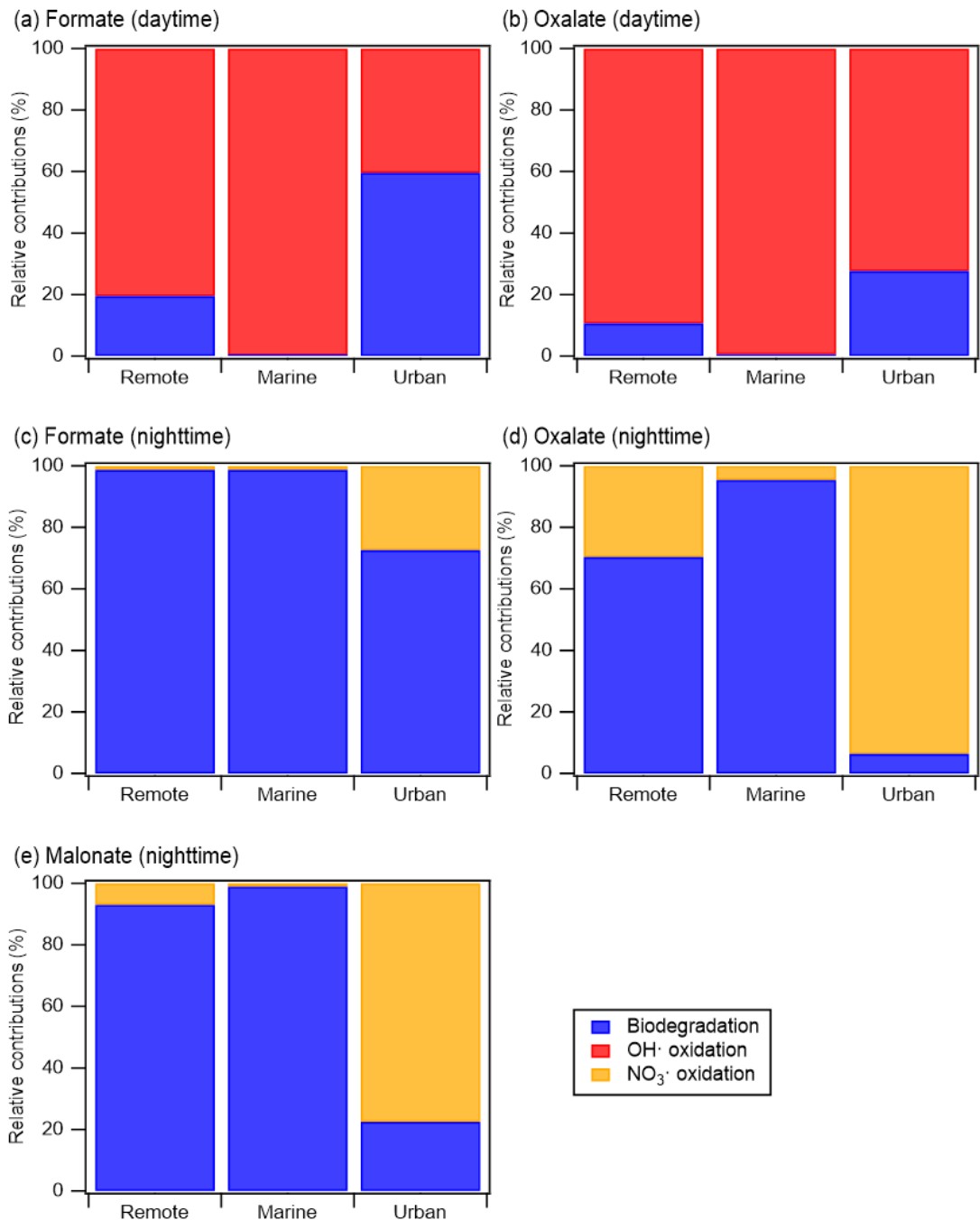

603

**Figure 5.** Predicted relative contributions of bacterial activity and chemical reaction (i.e., ·OH oxidation (daytime) and NO$_3$· (nighttime)) to the degradation of organic compounds in the aqueous phase in remote, marine, and urban areas. This figure is based on estimated loss rates shown in Table S7.

**4. Summary and implications**

In this study, we investigated how cloud water pH and exposure to solar radiation
impact the survival and energetic metabolism of bacteria and their ability to biodegrade organic
acids in clouds. Laboratory experiments were performed using artificial solar radiation and
artificial cloud water that mimicked the pH and composition of cloud water previously
collected in South China, which is a region with fairly acidic cloud water (pH 3 to 5.9). Using
two *E. hormaechei* strains that were isolated from ambient air in Hong Kong, we observed that
the energetic metabolism and survival of both strains depended on the artificial cloud water
pH. Low survival rates were observed for both strains at pH < 4 regardless whether the strains
were exposed to light. At pH 4 to 5, the energetic metabolism and survival of both strains were
only negatively impacted when they were exposed to light. In contrast, there were minimal
effects on the energetic metabolism and survival of both strains when they were exposed to
simulated sunlight at pH > 5. In addition, the biodegradation of organic acids depended on the
presence (or absence) of light and the artificial cloud water pH. The measured biodegradation
rates were around $10^{-19}$ to $10^{-18}$ mol cell$^{-1}$ s$^{-1}$, which were on the same order of magnitude as
the bacterial strains isolated from cloud water and implemented into cloud models (Vaitilingom
et al., 2010; Vaïtilingom et al., 2011; Fankhauser et al., 2019). Our analysis indicated that the
organic acid, cloud water pH, radical oxidant concentration, and the time of day will impact
the relative contributions of bacterial activity vs. $\cdot$OH/NO$_3\cdot$ chemistry in the aqueous phase.
This study has two important implications for our understanding of bacteria in clouds.
First, this study underscores the importance of accounting for cloud water pH when simulating
cloud processes involving metabolically active bacteria in atmospheric models, including
microbiological-chemical interactions between live bacteria and organic matter. Results from
this study imply the cloud water pH will impact the bacteria's ability to survive and thrive in
during the daytime and/or nighttime. The pH of cloud water typically lies between 3 and 6 (Pye
et al., 2020). Regions with high inputs of sulfuric acid and/or nitric acid combined with low
inputs of ammonia, dust, and sea salt, such as South China, will have moderately acidic to
highly acidic cloud water (Li et al., 2020; Pye et al., 2020; Shah et al., 2020; Qu and Han,
2021). Most of the bacteria in the atmosphere are neutrophiles that generally survive and thrive
in less acidic environments. Hence, even though our study focuses on two *Enterobacter* strains,

we hypothesize that cloud water pH will also affect the ability of other neutrophilic bacteria species to survive and remain metabolically active. Second, results from this study imply that it is important to consider the potential synergistic negative impacts that different stressors have on the survival and microbial activity of bacteria in clouds. Much of our current knowledge on the effect of different stressors (osmotic shocks, freeze-thaw cycles, and exposure to light and $H_2O_2$) on the survival of bacteria in clouds originate from a previous study by Joly et al. (2015) who investigated the impacts of these four stressors individually. However, as demonstrated in this study, when combined together, some stressors (in this case, cloud water pH and exposure to sunlight) can have synergistic negative impacts on the survival and microbial activity of bacteria in clouds.

While this study builds on our existing knowledge of how different stressors will impact the survival and energetic metabolism of bacteria and their ability to biodegrade organic matter in clouds, there are a number of caveats that should be noted. First, we were limited to using bacterial strains isolated from ambient air in this study due to the unavailability of bacteria isolates from cloud water in South China. Thus, if available, this work could be extended to bacteria isolates from cloud water in South China in the future to determine the pH conditions at which these isolates can survive and participate in microbiological-chemical interactions during the daytime and/or nighttime. The effect of cloud water pH on bacteria species that are reportedly common in cloud water (e.g., *Sphingomonadales*, *Rhodospirillales*, *Rhizobiales*, *Burkholderiales*, *Pseudomonadales* (Vaïtilingom et al., 2012; Zhu et al., 2018; Peng et al., 2019)) should also be investigated. Second, all the experiments in this study were conducted at 25 °C, which may be more representative of warmer regions during the summer (e.g., Hong Kong and parts of South China). Several studies have reported slower biodegradation rates at lower temperatures (Ariya et al., 2002; Vaitilingom et al., 2010; Husárová et al., 2011; Vaïtilingom et al., 2011), which suggest that cloud water temperature may influence the survival and energetic metabolism of bacteria. Third, the photon intensity in the photoreactor was kept constant in all the experiments. However, sunlight intensity will change throughout the day in the atmosphere. Fourth, this study does not consider how the presence of aqueous-phase oxidants (e.g., ·OH in the daytime, $NO_3$· in the nighttime) will impact the survival and

energetic metabolism of bacteria in clouds. Hence, the effects of temperature, light intensity, and oxidants on the impact the survival and energetic metabolism of bacteria and their ability to biodegrade organic matter in clouds should be investigated in future studies.

**Data availability:** The data used in this publication is available to the community and can be accessed on request to the corresponding author (theodora.nah@cityu.edu.hk), or at: https://doi.org/10.5281/zenodo.7045510 (Liu et al., 2022).

**Author contributions:** Y.L., P.L., and T.N. designed the study. Y.L. conducted the experiments. Y.L., C.K.L., and Z.S. performed the data analysis. Y.L. and T.N. wrote the manuscript with contributions from all co-authors.

**Competing interests:** One of the authors is a member of the editorial board of *Atmospheric Chemistry and Physics*. The peer-review process was guided by an independent editor, and the authors also have no other competing interests to declare.

**Acknowledgements:** This work was supported by the National Natural Science Foundation of China (project number R-BTC7801) and the Research Grants Council of Hong Kong (project number 11303720).

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
