# Peer review of "Effects of pH and light exposure on the survival of bacteria and their ability to biodegrade organic compounds in clouds: Implications for microbial activity in acidic cloud water"

_Atmospheric Chemistry and Physics, 2022_

## Author Comment (AC1)

We thank the referees for their careful reading and the detailed comments. The responses to the comments of the two referees in our direct reply (shown below) and within the revised manuscript (see marked copy) are provided below. The pages and lines indicated below correspond to those in the marked copy.

**Response to Referee 1 (Referees' comments are italicized)**

1. Referee comment: "*Pg 4, ln 122. Please elaborate on how and where the aerosol samples, from which the bacterial strains were isolated, were collected. For example, is the altitude or temperature of the sampling location known?*"

**Author response:** We have added the following information to the revised manuscript:

**Page 5, line 167: "Two new strains (B0910 and pf0910) belonging to *Enterobacter* species were isolated by exposing nutrient agar plates to ambient air in an urban environment (22.3360° N, 114.1732° E) at the height of 50 m above sea level during the summer season (~22 °C) in Hong Kong."**

2. Referee comment: "*Please comment on whether the bacteria were metabolically active in the atmosphere or rather on the possibility that they might have been dormant. What implications does this have for the results?*"

**Author response:** Although the atmosphere is historically considered to be a hostile environment for microbes, airborne metabolically active bacteria are present in the atmosphere. Culture-based and culture-independent analyses previously performed on bacteria isolated from cloud water showed that a fraction of these bacteria species are metabolically active. In addition, previous studies have reported that some of the culturable bacteria species isolated from cloud water can biodegrade atmospheric organic compounds, which provides further evidence that there are metabolically active bacteria in cloud water. We expect results from our study to provide new insights into how cloud water pH and exposure to solar radiation affect the survival and energetic metabolism of metabolically active bacteria and their ability to biodegrade organic compounds in clouds, which has important implications for our understanding of metabolically active bacteria in clouds.

The following information has been added into the revised manuscript to emphasize that metabolically active bacteria are present in the atmosphere:

**Page 2, line 71: "Airborne bacteria are comprised of both dead or dormant cells and metabolically active cells. Previous culture-based and culture-independent analyses of bacteria isolated from cloud water have shown that some of these bacteria species are metabolically active (Amato et al., 2007; Krumins et al., 2014; Amato et al., 2019). Previous studies have reported that the degradation of organic compounds as a result of microbiological-chemical interactions between live bacteria and organic matter can play an important role in influencing the carbon budget in clouds, which will have important atmospheric and climate implications (Delort et al., 2010; Vaitilingom et al., 2010; Vaitilingom et al., 2013; Ervens and Amato, 2020)."**

3. Referee comment: "*These experiments were conducted in 5 mL volumes based on Methods section 2.2 (pg 6, ln 176). Multiple points in the manuscript (e.g., pg 6, ln 179; pg 7, ln 196; pg 8, ln 226) mentioned aliquots of the sample being removed for analysis. What volume was removed, and would this in anyway bias the results? Were the test tubes stirred over the course of the experiments, and if not, could this impact the data?*"

**Author response:** During each experiment, the photoreactor's rotating rack held sixteen quartz tubes, all of which contained 5 mL of the solution comprised of bacteria cells suspended in artificial cloud water solution. We pipetted 100 µL of sample at different time points for colony forming unit (CFU) counts to determine the culturable bacterial cell concentrations (and calculate the bacteria survival rates), and 20 µL of sample at different time points for ADP/ATP measurements. In experiments aimed at investigating the biodegradation of organic acids, we pipetted 0.6 mL of sample from different quartz tubes at different time points for IC analysis. The pipetted 100 µL, 20 µL and 0.6 mL volumes accounts for 2%, 0.4%, and 12% for the initial volume of solution in each quartz tube (5 mL), respectively. Since small volumes of samples were pipetted from different tubes for these analyses over the course of these experiments, we do not expect the volumes removed to bias the results. In experiments aimed at identifying water-insoluble and water-soluble biological material and organic compounds, we removed two tubes at different time points and used all the solutions in the two tubes (for a total of 10 mL of sample) for UPLC-MS measurements. While the quartz tubes were not stirred over the course of the experiments, the high reproducibility of our results from these experiments (which were performed in triplicate) suggests that the samples in the quartz tubes were likely homogeneous. Thus, we do not expect our experimental protocols to impact our data.

4. Referee comment: "*Is there evidence that bacteria were actively metabolizing in the acidity and light experiments if the decay of organics in the cloud water mimic could not be observed? Have any live/dead staining before and after light exposure been performed?*"

**Author response:** There are two observations that provide evidence that the bacterial cells were metabolically active during the carboxylic acid biodegradation experiments. First, the measured ADP/ATP ratios, which can be seen as a measure of the bacterial cellular energy state, strongly suggested that the bacterial cells were metabolically active. A live cell typically has a ADP/ATP ratio of around 0.25, while a dead cell will have a ADP/ATP ratio of around 6 (Koutny et al., 2006). Figure S13 shows that the initial ADP/ATP ratios at reaction time 0 min were close to 0.25. The ADP/ATP ratio did not increase over the course of the experiment under dark conditions at pH 4.3, and under illuminated and dark conditions at pH 5.9. Although the ADP/ATP ratios increased over the course of the experiment under illuminated conditions at pH 4.3, the final ADP/ATP ratios (around 0.30) were substantially lower than 6. Second, aliquots (100 µL) of the solutions were taken at every hour over 12 hours for CFU counts on LB agar at 37 °C for 16 hours to determine the culturable bacterial cell concentrations. Since we were able to grow the bacterial cells on the agar, this indicated that the bacterial cells were metabolically active. We did not stain the cells before and after the light exposure since we performed cell culture experiments to determine the fraction of live vs. dead cells in our study.

5. Referee comment: "*Pg 8, lns 220-222. While citations are provided to support the claim that the concentration ratio between chemicals and cells rather than absolute values are important*

*for degradation rates, the latter two both reference the first one (Vaïtilingom et al. 2010). In that case, it is for a different bacterium and for a single carbon source. Can their assumption be extended to these experiments given the discrepancies?"*

**Author response:** We acknowledge that the referee raised a valid point. We were unable to find other studies that performed experiments with different bacteria strains incubated in the presence of multiple carbon sources to support the claim that the concentration ratio between chemical compounds and bacterial cells rather than absolute values are important for biodegradation rates can be applied to experiments utilizing different bacteria strains and for multiple carbon sources. Hence, we have revised the manuscript to be more circumspect about the validity of this assumption:

**Page 8, line 258: "Due to the detection limits of the IC system used to measure the organic acids, the concentration for each organic acid was set to 50 µM (Table S2), which is around 10 times higher than the concentrations typically measured in cloud water. The concentrations of inorganic ions in the artificial cloud water were also increased by 10 times. Vaitilingom et al. (2010) previously reported that the same biodegradation rates will be obtained as long as the concentration ratio of the chemical compounds to bacterial cells is constant. However, the authors drew this conclusion based on experiments performed using a *Pseudomonas graminis* bacterial strain incubated in the presence of a single organic compound as the carbon source. At present, it is unclear whether this conclusion can be extrapolated to other bacteria species incubated in the presence of multiple organic compounds, and this warrants further study. Nevertheless, we made the same assumption (i.e., the same biodegradation rates will be obtained as long as the concentration ratio of the chemical compounds to bacterial cells is constant) as was done in previous studies that investigated the biodegradation of multiple organic compounds by different bacteria species (Vaïtilingom et al., 2011; Jaber et al., 2020; Jaber et al., 2021). Hence, the bacteria concentration used was set to $10^6$ cells mL$^{-1}$ to maintain the same concentration ratio of the organic acids to bacterial cells."**

6. Referee comment: "*What were the "simple" (pg 17, ln 410) calculations used to create Figure 5? Are they merely ratios of the degradation rates or were they based on a model? Either way, what are the assumptions used and the limitations of that approach?*"

**Author response:** We refer the referee to Section S5, which describes the simple kinetic analysis we performed using the biodegradation rates that we measured in this study. We have also added discussions on the assumptions used and the limitations of our approach in the revised manuscript:

**Page 22, line 609: "A simple kinetic analysis was performed to identify the factors that will impact the relative contributions of bacterial activity vs. ·OH/NO₃· chemistry in cloud water during the daytime and nighttime. Details of the calculations performed in this kinetic analysis can be found in Section S5. Our approach of considering daytime and nighttime processes separately was different from the approach used by previous studies, which determined the relative contributions of bacterial activity and chemical reactions on the degradation of organic compounds by only considering dark biodegradation**

processes and ·OH photochemical reactions (Vaïtilingom et al., 2011; Jaber et al., 2020; Jaber et al., 2021). Here, biodegradation rates that were measured under illuminated conditions were used for the daytime scenario, while biodegradation rates that were measured under dark conditions were used for the nighttime scenario. We used the average of biodegradation rates measured for the two strains for our calculations. Formate, oxalate, and malonate were chosen for our analysis since their ·OH and NO$_3$· reaction rate constants were available in the literature. ·OH and NO$_3$· are the main tropospheric aqueous-phase free radicals during the daytime and nighttime, respectively (Herrmann et al., 2010). The average measured biodegradation rates of formate, oxalate, and malonate were first converted to biodegradation rate constants. These biodegradation rate constants and the corresponding ·OH and NO$_3$· reaction rate constants provided by the literature (Table 1) were subsequently used for calculations of the biodegradation rates and chemical reaction rates in cloud water (Section S5). A bacteria concentration of $8 \times 10^7$ cell L$^{-1}$ was assumed in our calculations for the daytime scenario at pH ~5 and the nighttime scenarios at pH ~4 and ~5, which was the same bacteria concentration used in previous studies and represented the highest estimate of actual live bacteria concentrations (i.e., 100% of metabolically active cells) (Vaïtilingom et al., 2011; Jaber et al., 2020; Jaber et al., 2021). Based on our investigations of the survival and energetic metabolism of bacteria under illuminated conditions at pH 4 to 5 (Figure 1), we expect the bacteria concentrations to gradually decrease for the daytime scenario at pH ~4. Thus, for simplicity, we assumed a lower bacteria concentration in our calculations for the daytime scenario at pH ~4, whereby we multiplied the bacteria concentration of $8 \times 10^7$ cell L$^{-1}$ by a factor of 0.75. This factor was obtained by taking the average survival rates for the two strains from reaction time 0 to 12 hour in our experiments conducted under illuminated conditions at pH 4.3 (Figure S13). The rates of oxidation by ·OH and NO$_3$· chemical reactions will depend on their respective concentrations. Hence, we used the average ·OH and NO$_3$· concentrations reported by Herrmann et al. (2010) for remote, marine, and urban environments in our calculations (Table S6) (Herrmann et al., 2010)."

Page 24, line 669: "Our simple kinetic analysis indicated that the organic acid, cloud water pH, radical oxidant concentration, and time of day (i.e., daytime vs. nighttime) will impact the relative contributions of bacterial activity vs. ·OH/NO$_3$· chemistry in the aqueous phase. However, there are a number of caveats that should be noted. First, the biodegradation rates used in this analysis were from experiments conducted at 25 °C, which may be more representative of warmer regions during the summer (e.g., Hong Kong and parts of South China). Slower biodegradation rates will likely be measured at lower temperatures (Ariya et al., 2002; Vaitilingom et al., 2010; Husárová et al., 2011; Vaïtilingom et al., 2011), which will impact the relative contributions of bacterial activity vs. ·OH/NO$_3$· chemistry. Second, our analysis did not account for how the presence of aqueous-phase oxidants (e.g., ·OH in the daytime, NO$_3$· in the nighttime) will impact the survival and energetic metabolism of bacteria, which in turn will impact the relative contributions of bacterial activity vs. ·OH/NO$_3$· chemistry. Third, our analysis did not account for the physical separation of cloud droplets containing bacteria cells from cell-free cloud droplets. Only a small fraction of cloud droplets will contain metabolically

active bacteria cells, and the bacterial metabolism cannot affect the composition of organic acids in cell-free cloud droplets (Fankhauser et al., 2019; Khaled et al., 2021). Hence, only $\cdot OH/NO_3\cdot$ chemistry will govern the degradation of organic acids in cell-free droplets. Consequently, not accounting for the physical separation of cloud droplets containing bacteria cells from cell-free cloud droplets will result in an overestimation of the contribution of bacterial activity to the biodegradation of organic compounds (Fankhauser et al., 2019; Khaled et al., 2021). Fourth, our analysis only considers biodegradation and chemical reactions occurring in the aqueous phase and ignores gas-aqueous phase exchanges and gas-phase chemical reactions. Nah et al. (2018) previously showed that the gas-aqueous phase partitioning of organic acids will depend on the organic acid's Henry's law constant and acid dissociation constants, liquid water concentration, temperature, and pH (Section S6). Figure S14 shows that a significant fraction of formic acid will be in the gas phase at pH 4 and 5 under cloud water conditions, whereas all of oxalic acid, malonic acid, and maleic acid will be in the aqueous phase at pH 4 and 5 under cloud water conditions. This suggests that gas-phase chemical reactions will likely play an important role in consuming formic acid, whereas the consumption of oxalic acid, malonic acid, and maleic acid will likely mainly be through bacterial activity and chemical reactions in the aqueous phase. Quantifying the exact contributions of aqueous-phase bacterial activity vs. aqueous-phase $\cdot OH/NO_3\cdot$ chemistry vs. gas-phase $\cdot OH/NO_3\cdot$ chemistry under different cloud water pH conditions will require a multi-phase box model similar to the one used by Khaled et al. (2021). This is beyond the scope of the current study but can be a subject of future studies."

7. Referee comment: "*Furthermore, although Fankhauser et al. 2019 is referenced (pg 16, ln 379; pg 21, ln 478), their main finding was not acknowledged. They concluded that metabolically active microorganisms in the atmosphere are physically separated from the majority of atmospheric water (and thus organics), and that bacterial metabolism should not significantly affect the total organic content. This seems to directly contradict the finding that bacterial metabolism can be competitive with chemical oxidation. How can this be reconciled?*"

**Author response:** We agree with the referee that the main findings of Fankhauser et al. (2019) need to be discussed. We acknowledge that neglecting the physical separation of cloud droplets containing bacteria cells from cell-free cloud droplets will lead to an overestimation of the potential impacts of bacterial metabolism. Thus, we likely overstated the competitiveness of bacterial metabolism in our original manuscript since we did not consider the physical separation of cloud droplets containing bacteria cells from cell-free cloud droplets. We have revised the manuscript to be more circumspect about the competitiveness of bacterial metabolism vs. chemical reactions in degrading organic acids in the aqueous phase. We refer the referee to our response to their previous comment, which also shows the changes we made in the revised manuscript to address this comment.

8. Referee comment: "*Technical comments: Please define ADP and ATP.*"

**Author response:** We have defined ADP and ATP in the revised manuscript:

**Page 7, line 225: "The adenosine diphosphate/adenosine triphosphate (ADP/ATP) ratios**

**were measured using an assay kit (EnzyLightTM, BioAssay Systems) and a biolumineter (SpectraMax M2e) to determine changes in the bacteria energetic metabolism."**

**Response to Referee 2 (Referees' comments are italicized)**

1. Referee comment: "*The manuscript is written in correct language, but there is a general lack of precision in the statements (MSA is not a carboxylic acid for instance), important references are missing or misinterpreted (such as Khaled et al. (2021), where the necessity to account for the multiphase and heterogeneous aspects of clouds is emphasized, or the (inexact) statement that biodiversity in clouds was only investigated so far through culture methods L60. In addition, in several places the interpretations could be expanded and contextualized.*"

**Author response:** We have addressed these concerns in the revised manuscript. Please refer to our response to the comments made by both referees.

2. Referee comment: "*I had difficulties to figure out what was the rationales behind some aspects of the study that not seem to bring additional information, and that are not discussed into context, such as ADP/ATP data and the list of organics released during cell lysis. The latter in particular appears totally disconnected from the rest of the study, barely out of subject, while it could be presented and discussed as a source of organics to cloud water, and/or biomarkers of cell lysis. Are these compounds indeed found in natural cloud water? Could these be used by the remaining active cells as substrates? Could these be used as biomarkers of damages to cells in such environments? About ADP and ATP, could these be used as proxies to evaluate survival and biodegradation rates in natural situations? Does this variable provide additional information here compared to cultures regarding survival?*"

**Author response:** We thank the referee for his helpful suggestions on how to better connect our discussions regarding ADP/ATP and the organic compounds released during cell lysis to the other results presented the manuscript. The following changes were made to the revised manuscript:

**Page 17, line 473: "These detected compounds indicated that bacterial cell lysis could be a source for carbon in cloud water. Many of the compound classes detected in this study have previously been measured in atmospheric cloud water. For instance, large abundances of peptide-like compounds and lipid-like compounds have been measured in cloud water from Puy de Dôme (Bianco et al., 2018; Bianco et al., 2019), which is consistent with the detection of large abundances of compounds assigned to the peptide and lipid compound classes in this study. This suggested that peptide-like and lipid-like compounds could be used as biomarkers to evaluate bacterial contributions to atmospheric samples. Previous studies have used fatty acids, which are integral building blocks of lipids, in atmospheric samples as biomarkers for characterizing and quantifying bacteria, and assessing the atmospheric transport of bacteria (Kawamura et al., 2003; Lee et al., 2004; Tyagi et al., 2015). While this study shows that bacterial cell lysis will release large quantities of peptide-like and lipid-like compounds, using these compounds as biomarkers for bacterial cell lysis in atmospheric samples will likely be complex as the**

concentrations of these compounds will likely change with time. This is because peptide-like and lipid-like compounds will undergo chemical and biological transformations after they have been released during cell lysis, which will impact their concentrations in atmospheric samples. Amino acids, which are building blocks of peptides, are known to undergo chemical reactions with oxidants in cloud water, (Bianco et al., 2016). In addition, peptide-like and lipid-like compounds can be produced and/or consumed by cloud microorganisms to maintain their metabolism (Bianco et al., 2019; Jaber et al., 2021)."

**Page 9, line 305:** "Figure 1 shows the survival rates and ADP/ATP ratios of the *E. hormaechei* B0910 and *E. hormaechei* pf0910 strains over time under illuminated and dark conditions at different artificial cloud water pH. The ADP/ATP ratio is used as an indicator of the bacteria's metabolic activity and survival rate in this study. Growing cells usually maintain a constant ADP/ATP ratio because whenever there is a decrease in intracellular ATP production, its degradation product ADP will be resynthesized to form ATP to maintain intracellular ATP concentrations (Koutny et al., 2006; Guan and Liu, 2020). In contrast, when there is a disruption in the metabolism of ATP production, ATP cannot be resynthesized from ADP even though ATP is still converted to ADP, which will cause the ADP/ATP ratio to increase (Koutny et al., 2006; Guan and Liu, 2020)"

3. Referee comment: "*The genomes of the bacteria investigated were sequenced, and such data could provide useful information to interpret the data. Nevertheless, genomes have not been exploited at all in this study. Looking for relevant enzymes and functions in the genomes (pH homeostasis, internalization and use of carbon substrates, etc) and discussing them would definitely strengthen the paper regarding the biological aspects of cloud microbiology.*"

**Author response:** As requested, we have provided more discussion about the bacteria genomes and their roles in the revised manuscript. We moved our previous discussion on the possible enzymes, functions, and mechanisms associated with organic acid biodegradation by the two bacterial strains from the Supplementary Information to the revised main manuscript. We also added a discussion about pH homeostasis to the revised manuscript. The following changes were made to the revised manuscript:

**Page 12, line 354:** "Our genomic analysis revealed that the two strains have genes encoding a F1F0-type ATP synthase, which can export protons from their cytoplasm to cope with pH stress (Krulwich et al., 2011). In addition, genes encoding potassium transporters, which may be involved in pH homeostasis (i.e., both Kup-type low-affinity and Kdp-type high-affinity potassium transporters) (Brzoska et al., 2022) were found in the genome of both strains (Table S3)."

**Table S3. Genes involved in the pH homeostasis in the two *E. hormaechei* strains.**

| Transporters | Protein subunits | *E. hormaechei B0910* | *E. hormaechei pf0910* |
|---|---|---|---|
| | | CDS | CDS |
| **F1F0-type ATP** | **Subunit a, AtpB** | **MOG78_16595** | **MMW20_13045** |

| synthase | Subunit c, AtpE | MOG78_16590 | MMW20_13050 |
| --- | --- | --- | --- |
| | Subunit b, AtpF | MOG78_16585 | MMW20_13055 |
| | Subunit delta, AtpH | MOG78_16580 | MMW20_13060 |
| | Subunit alpha, AtpA | MOG78_16575 | MMW20_13065 |
| | Subunit gamma, AtpG | MOG78_16570 | MMW20_13070 |
| | Subunit beta, AtpD | MOG78_16565 | MMW20_13075 |
| | Subunit epsilon, AtpC | MOG78_16560 | MMW20_13080 |
| Kdp-type high-affinity potassium transporter | Potassium-binding ATPase subunit KdpA | MOG78_10080 | MMW20_19865 |
| | Potassium-binding ATPase subunit KdpB | MOG78_10085 | MMW20_19860 |
| | Potassium-binding ATPase subunit KdpC | MOG78_10090 | MMW20_19855 |
| | Potassium-binding ATPase subunit KdpF | MOG78_10075 | Gene sequence found but CDS is not annotated. (Chromosome genome nucleotide position: 3800683-3800772) |
| Kup-type low-affinity potassium transporter | Kup | MOG78_16640 | MMW20_13000 |

**Page 18, line 514: "Table S5 summarizes the enzymes or metabolic pathways related to the biodegradation of organic acids in the two strains. Genes encoding formate dehydrogenases were identified in both genomes, which is consistent with the observed formate biodegradation. However, no known genes for oxalic acid biodegradation (Liu et al., 2021) were found in the genomes of both strains, which suggested the presence of yet to be characterized pathways that catalyzed the biodegradation. Interestingly, a protein with Cupin 2 domain was found in both genomes. The Cupin superfamily consists of a diverse range of enzymes including oxalate oxidase and oxalate decarboxylase that can biodegrade oxalic acid (Burrell et al., 2007).**

**Only the *E. hormaechei* B0910 strain was observed to biodegrade malonic acid. Interestingly, the malonyl-CoA-acyl carrier transcacylase observed in the *E. hormaechei* pf0910 strain seems to be a fusion protein, which may render it ineffective in utilizing malonic acid. Although no gene encoding maleate isomerase was identified in the genomes**

of both strains, the maleic acid biodegradation observed can be attributed to the activity of other enzymes with broad substrates specificity (Hatakeyama et al., 2000). The genes encoding for the small and large protein subunits that together form the 3-isopropylmalate dehydratase, the enzyme that isomerizes 2-isopropylmalate to 3-isopropylmalate, were found in both the *Enterobacter* strains. The small and large protein subunits of this enzyme are homologous to the small (51% amino acid identity) and large (59% amino acid identity) protein subunit constituents of maleate hydratase (HbzIJ) from Pseudomonas alcaligenes NCIMB 9867 that converts maleate to D-malate (Liu et al., 2015). Given the high protein homology, we speculate that the 3-isopropylmalate dehydratase in the *Enterobacter* strains may have a broader substrate specificity than known and it may be able to biodegrade maleate.

The lack of biodegradation of acetic acid, MSA, and glutaric acid in the experiments could be partly explained by the genomic information. Both strains have genes that encode enzymes involved in the biodegradation (Table S5) and associated uptake transporters (i.e., acetate permease (ActP) and succinate-acetate/proton symporter (SatP)) of acetic acid. The lack of the corresponding biodegradation in the experiments could be due to the low uptake of acetic acid by cells as ActP functions to scavenge low concentrations of the compound (Gimenez et al., 2003) while SatP could be inhibited by formic acid found in the cloud water medium (Sá-Pessoa et al., 2013). Genes encoding the two-component alkanesulfonate monooxygenase for MSA biodegradation were found in both strains, but they were likely not expressed as sulfur was not deficient in the cloud water medium (Kahnert et al., 2000; Eichhorn and Leisinger, 2001), which is consistent with the absence of MSA biodegradation in the experiments. While genes encoding succinate-semialdehyde dehydrogenase/glutarate-semialdehyde dehydrogenase, which display a reversible conversion between glutarate-semialdehyde and glutarate in the KEGG database (Kanehisa et al., 2022), were found in both strains, to the best of our knowledge there is no report of experimental results confirming that the reaction can go in the reverse direction from glutarate to glutarate-semialdehyde. In addition, a study of glutaric semialdehyde dehydrogenase reported the irreversible nature of the catalysis of glutarate semialdehyde to glutarate (Ichihara and Ichihara, 1961). Thus, it is not surprising that glutarate biodegradation was not observed for the two strains."

4. Referee comment: "*It is not clear at what time points the biodegradation rates were calculated from experiments. Were [cell]experiment adjusted for accounting for the decrease of survival? In this regard, it is indicated (line 428) that "a constant bacteria concentration of $8 \times 10^7$ cell L$^{-1}$ was assumed in our calculations". First, it has to be clear that this number is the highest estimate of the actual active cell concentrations (i.e. 100% of active cells), as these correspond to total cell numbers in the references cited, and second: it appears odd to consider active cells constant in acidic clouds while at the same time presenting data showing that survival and activity are affected. This aspect would need at least a bit of clarification and discussion.*"

**Author response:** The biodegradation rates were determined from fits to the organic acid decays from reaction time 0 to 12 hours during each experiment (see Section 2.4). The referee

is correct in stating that the assumed bacteria concentration ($8 \times 10^7$ cell L$^{-1}$) in our calculations of the biodegradation rates in cloud water is the highest estimate of the actual bacteria concentrations (i.e., 100% active cells). There were no noticeable changes in the bacteria survival rates under dark and illuminated conditions at pH 5.9, and under dark conditions at pH 4.3. Thus, we did not adjust the bacteria cell concentrations for these three conditions, and we assumed a constant bacteria cell concentration of $8 \times 10^7$ cell L$^{-1}$ in our calculations for the daytime scenario at pH ~5 and the nighttime scenarios at pH ~4 and ~5. Since there were noticeable changes in the bacteria survival rates under illuminated conditions at pH 4.3, we accounted for the lower bacteria survival rates in our calculations for the daytime scenario at pH ~4 by multiplying the bacteria cell concentration of $8 \times 10^7$ cell L$^{-1}$ by a factor of 0.75. This factor was obtained by taking the average survival rates for the two strains from reaction time 0 to 12 hour in (Figure S13). This information, which was not included in the original manuscript, will be added into the revised manuscript. The following changes were made to the revised manuscript:

**Page 20, line 565: "Figure 4 summarizes the measured biodegradation rates of the organic acids for the two strains under dark and illuminated conditions at pH 4.3 and pH 5.9. These biodegradation rates were determined from fits to the decays of the organic acids from reaction time 0 to 12 hour in each experiment (Section 2.4)."**

**Page 22, line 627: "A bacteria concentration of $8 \times 10^7$ cell L$^{-1}$ was assumed in our calculations for the daytime scenario at pH ~5 and the nighttime scenarios at pH ~4 and ~5, which was the same bacteria concentration used in previous studies and represented the highest estimate of actual live bacteria concentrations (i.e., 100% of metabolically active cells) (Vaïtilingom et al., 2011; Jaber et al., 2020; Jaber et al., 2021). Based on our investigations of the survival and energetic metabolism of bacteria under illuminated conditions at pH 4 to 5 (Figure 1), we expect the bacteria concentrations to gradually decrease for the daytime scenario at pH ~4. Thus, for simplicity, we assumed a lower bacteria concentration in our calculations for the daytime scenario at pH ~4, whereby we multiplied the bacteria concentration of $8 \times 10^7$ cell L$^{-1}$ by a factor of 0.75. This factor was obtained by taking the average survival rates for the two strains from reaction time 0 to 12 hour in in our experiments conducted under illuminated conditions at pH 4.3 (Figure S13)."**

5. Referee comment: "*Besides survival, there are likely other aspects related with pH that could be considered/discussed: how it modifies the form of the organic compounds studied (pKa) and so their biological availability? Can this impact the solubilization and volatilization of the organic compounds in clouds, and so their biological use in cloud droplets? Additionally, different organic compounds were mixed together in the incubation medium. Was a prioritization observed, i.e. were some substrates preferentially used over others? Your work could provide valuable information here.*"

**Author response:** As requested, we have added discussions about other pH-related aspects to the revised manuscript. Based on experimental results, formic acid appears to be preferentially consumed by the two bacteria strains used in our study. However, we do not know if this observation can be extrapolated to other bacteria species. The following changes were made in

the revised manuscript:

**Page 18, line 502: "The biodegradation of seven organic acids (i.e., formic acid, acetic acid, oxalic acid, maleic acid, malonic acid, glutaric acid and MSA) that were mixed together were measured under dark and illuminated conditions at pH 4.3 and pH 5.9. Only some of the seven organic acids were biodegraded by the two strains. Based on our experimental conditions (liquid water content $\approx 10^{12}$ μg m$^{-3}$, the density of water) and the organic acids' Henry's law constants (Table S8), these organic acids will be in the aqueous phase and are not expected to volatilize during these experiments. Thus, the observed decays were due to bacterial metabolism."**

**Page 24, line 689: "Fourth, our analysis only considers biodegradation and chemical reactions occurring in the aqueous phase and ignores gas-aqueous phase exchanges and gas-phase chemical reactions. Nah et al. (2018) previously showed that the gas-aqueous phase partitioning of organic acids will depend on the organic acid's Henry's law constant and acid dissociation constants, liquid water concentration, temperature, and pH (Section S6). Figure S14 shows that a significant fraction of formic acid will be in the gas phase at pH 4 and 5 under cloud water conditions, whereas all of oxalic acid, malonic acid, and maleic acid will be in the aqueous phase at pH 4 and 5 under cloud water conditions. This suggests that gas-phase chemical reactions will likely play an important role in consuming formic acid, whereas the consumption of oxalic acid, malonic acid, and maleic acid will likely mainly be through bacterial activity and chemical reactions in the aqueous phase."**

**SI, Page 28, line 415: "Section S6. Gas-aqueous phase partitioning of monocarboxylic and dicarboxylic acids**

Meskhidze et al. (2003) and Guo et al. (2016) previously introduced the concept of "S curves", which describe how the pH of the aqueous phase affects the gas-aqueous partitioning of acidic and basic species. It is assumed that the equilibrium between gas and aqueous phases involves the dissolution of the acidic/basic species into the aqueous phase, followed by the dissociation of the dissolved species. Assuming unity activity coefficients, for monocarboxylic acids (HA, e.g., formic acid), the pH-dependence of the molar fraction of HA in the aqueous phase ($\varepsilon(HA(aq))$) is described by the following equation (Nah et al., 2018):

$$\varepsilon(HA(aq)) = \frac{H_{HA}WRT(10^{-pH} + K_{a1}) \times 0.987 \times 10^{-14}}{10^{-pH} + H_{HA}WRT(10^{-pH} + K_{a1}) \times 0.987 \times 10^{-14}}$$

where $W$ is liquid water concentration (μg m$^{-3}$), $H_{HA}$ (mole L$^{-1}$ atm$^{-1}$) is the Henry's law constants for monocarboxylic acid, $K_{a1}$ (mole L$^{-1}$) is the first acid dissociation constant, $R$ is the gas constant (8.314 m$^3$ Pa K$^{-1}$ mol$^{-1}$), and $T$ is temperature (K). The complete derivation for $\varepsilon(HA(aq))$ can be found in the SI of Guo et al. (2015).**

Assuming unity activity coefficients, for dicarboxylic acids (H2A, e.g., oxalic acid, malonic acid, and maleic acid), the pH-dependence of the molar fraction of H2A in the aqueous phase ($\varepsilon(H_2A(aq))$) can eventually be simplified to the following equation (Nah

**et al., 2018):**

$$\varepsilon(H_2A(aq)) \cong \frac{H_{H_2A}WRT(10^{-pH} + K_{a1}) \times 0.987 \times 10^{-14}}{10^{-pH} + H_{H_2A}WRT(10^{-pH} + K_{a1}) \times 0.987 \times 10^{-14}}$$

where $W$ is liquid water concentration (μg m$^{-3}$), $H_{H_2A}$(mole L$^{-1}$ atm$^{-1}$) is the Henry's law constants for monocarboxylic acid, $K_{a1}$ (mole L$^{-1}$) is the first acid dissociation constant, $R$ is the gas constant (8.314 m$^3$ Pa K$^{-1}$ mol$^{-1}$), and $T$ is temperature (K). The complete derivation for $\varepsilon(H_2A(aq))$ can be found in the SI of Nah et al. (2018), which also includes discussions of the assumptions made during the derivation process which will lead to the disappearance of the second acid dissociation constant ($K_{a2}$) term during the process of simplifying the equation."

[Figure]

**Figure S14.** Calculated pH-dependent molar fraction of formic acid in the aqueous phase ($\varepsilon(HA(aq))$)) and pH-dependent molar fractions of oxalic acid, malonic acid, and maleic acid in the aqueous phase ($\varepsilon(H_2A(aq))$)) under cloud water conditions (Section S6 and Table S8). A liquid water concentration of 10$^6$ μg m$^{-3}$ (Ervens et al., 2011) was assumed in these calculations. A significant fraction of formic acid will be in the gas phase at pH 4 and 5 under cloud water conditions, whereas all of the oxalic acid, malonic acid, and maleic acid will be in the aqueous phase at pH 4 and 5 under cloud water conditions (note that their values overlap one another at $\varepsilon(H_2A(aq)) = 1$). These differences were due primarily to the substantial differences in their water solubility (i.e., Henry's law constants) (Table S8).

**Table S8.** Acid dissociation constants and Henry's law coefficients at 25 °C used to generate $\varepsilon(HA(aq))$ and $\varepsilon(H_2A(aq))$ S curves in Figure S14

| Organic acid | First acid dissociation constant ($K_{a1}$) (mol L$^{-1}$) | Second acid dissociation constant ($K_{a2}$) (mol L$^{-1}$) | Henry's law constant ($H_{HA}$ or $H_{H_2A}$) (mol L$^{-1}$ atm$^{-1}$) |
|---|---|---|---|

| Formic acid | $1.78 \times 10^{-4}$ (Haynes, 2014) | Not applicable | $9.53 \times 10^{3}$ |
|---|---|---|---|
| Oxalic acid | $5.62 \times 10^{-2}$ (Haynes, 2014) | $1.55 \times 10^{-4}$ (Haynes, 2014) | $6.11 \times 10^{8}$ (Nah et al., 2018)[a] |
| Malonic acid | $1.48 \times 10^{-3}$ (Williams, 2022) | $2.04 \times 10^{-6}$ (Williams, 2022) | $3.85 \times 10^{10}$ (Compernolle and Müller, 2014) |
| Maleic acid | $1.26 \times 10^{-2}$ (Weast and Astle, 1981) | $8.51 \times 10^{-7}$ (Weast and Astle, 1981) | $1.42 \times 10^{10}$ (Lide and Frederikse, 1995) |

[a]**While we used the Henry's law coefficient provided by Nah et al. (2018), it should be noted that the authors obtained this value by taking the average of $H_{C_2H_2O_4}$ values provided by Clegg et al. (1996), Compernolle and Muller (2014) and Saxena and Hildemann (1996), and accounted for the effect of temperature using the equations provided by Sander (2015).**

6. Referee comment: "*Absence of statistics: Statistically significant differences are mentioned (L391 and elsewhere), but the tests used and the results are not specified; these should be.*"

**Author response:** We used the Student's t test in the statistical analysis. These results were shown in Figure S12 of Supplement Information in the previous manuscript. We have added information about the statistical analysis performed in the revised manuscript:

**Page 20, line 581: "For the effect of artificial cloud water pH on the biodegradation of organic acids by *E. hormaechei* B0910, the differences in the biodegradation rates were statistically significant for the four acids (Student's t test, $p$ value < 0.05). Conversely, the differences in the biodegradation rates of formate and oxalate as a result of light exposure were statistically significant at pH 5.9 (Student's t test, $p$ value < 0.05). For the effect of artificial cloud water pH on the biodegradation of organic acids by *E. hormaechei* pf0910, only the difference in the dark biodegradation of oxalate was statistically significant (Student's t test, $p$ value < 0.05). In contrast, light exposure reduced the formate biodegradation rates significantly at both pH 4.3 and pH 5.9 (Student's t test, $p$ value < 0.05), and the oxalate biodegradation rate significantly at pH 5.9 (Student's t test, $p$ value < 0.05)."**

**SI, Page 13, line 106: "Figure S12. Biodegradation rates of oxalate, maleate, and malonate by (a) *E. hormaechei* B0910 and (b) *E. hormaechei* pf0910 under light and dark conditions at pH 4.3 and pH 5.9. Error bars represent one standard deviation from the mean of biological triplicates. Statistical analysis was performed using the Student's t test (ns: not significant, \*: $p$ value < 0.05, \*\*: $p$ value < 0.01, \*\*\*: $p$ value < 0.001)."**

7. Referee comment: "*About modeling, the approach used is quite similar as early work regarding the evaluation of the impacts of biological activity on cloud chemistry (e.g. (Vaïtilingom et al., 2011). This basically consisted in comparing the biodegradation rates to radical chemistry rates. However, such a simplistic approach totally omits the main specificity of cloud water, i.e. its distribution in droplets. This is my main concern regarding this work. Yet*

*distribution in droplets has been shown to be a key factor in assessing the impacts of biological degradation in clouds (Khaled et al., 2021), particularly because not all droplet contain bacteria cells, and because water-air exchanges are critical in such systems. Therefore, the predicted relative contributions of bacteria and radicals to the loss of organic compounds in this work are ultimately valid only for the population of droplets containing bacteria cells, but not for the entire cloud. This should be clearly stated and if possible, this part of the work could be reevaluated.*"

**Author response:** Our main goal for performing this simple analysis is to demonstrate that the organic acid, cloud water pH, radical oxidant concentration, and time of day (i.e., daytime vs. nighttime) will impact the relative contributions of bacterial activity vs. $\cdot OH/NO_3\cdot$ chemistry in the aqueous phase. We agree with the referee that our approach is a little simplistic and omits details such as the physical separation of cloud droplets containing bacteria cells from cell-free cloud droplets and gas-aqueous exchanges. We have added a short discussion on the limitations of our approach in the revised manuscript. We acknowledge that neglecting the physical separation of cloud droplets containing bacteria cells from cell-free cloud droplets will lead to an overestimation of the potential impacts of bacterial metabolism. Thus, we likely overstated the competitiveness of bacterial metabolism in our original manuscript since we did not consider the physical separation of cloud droplets containing bacteria cells from cell-free cloud droplets. We have revised the manuscript to be more circumspect about the competitiveness of bacterial metabolism vs. chemical reactions in degrading organic acids in the aqueous phase. Our new calculations also showed that a significant fraction of formic acid will be in the gas phase at pH 4 and 5 under cloud water conditions, while all of oxalic acid, malonic acid, and maleic acid will be in the aqueous phase at pH 4 and 5 under cloud water conditions. This suggests that gas-phase chemical reactions will likely play an important role in consuming formic acid, and this is now clearly stated in the revised manuscript. The following changes have been made to the revised manuscript:

**Page 24, line 669: "Our simple kinetic analysis indicated that the organic acid, cloud water pH, radical oxidant concentration, and time of day (i.e., daytime vs. nighttime) will impact the relative contributions of bacterial activity vs. $\cdot OH/NO_3\cdot$ chemistry in the aqueous phase. However, there are a number of caveats that should be noted. First, the biodegradation rates used in this analysis were from experiments conducted at 25 °C, which may be more representative of warmer regions during the summer (e.g., Hong Kong and parts of South China). Slower biodegradation rates will likely be measured at lower temperatures (Ariya et al., 2002; Vaitilingom et al., 2010; Husárová et al., 2011; Vaïtilingom et al., 2011), which will impact the relative contributions of bacterial activity vs. $\cdot OH/NO_3\cdot$ chemistry. Second, our analysis did not account for how the presence of aqueous-phase oxidants (e.g., $\cdot OH$ in the daytime, $NO_3\cdot$ in the nighttime) will impact the survival and energetic metabolism of bacteria, which in turn will impact the relative contributions of bacterial activity vs. $\cdot OH/NO_3\cdot$ chemistry. Third, our analysis did not account for the physical separation of cloud droplets containing bacteria cells from cell-free cloud droplets. Only a small fraction of cloud droplets will contain metabolically active bacteria cells, and the bacterial metabolism cannot affect the composition of organic acids in cell-free cloud droplets (Fankhauser et al., 2019; Khaled et al., 2021).**

Hence, only ·OH/NO₃· chemistry will govern the degradation of organic acids in cell-free droplets. Consequently, not accounting for the physical separation of cloud droplets containing bacteria cells from cell-free cloud droplets will result in an overestimation of the overall contribution of bacterial activity to the biodegradation of organic compounds (Fankhauser et al., 2019; Khaled et al., 2021). Fourth, our analysis only considers biodegradation and chemical reactions occurring in the aqueous phase and ignores gas-aqueous phase exchanges and gas-phase chemical reactions. Nah et al. (2018) previously showed that the gas-aqueous phase partitioning of organic acids will depend on the organic acid's Henry's law constant and acid dissociation constants, liquid water concentration, temperature, and pH (Section S6). Figure S14 shows that a significant fraction of formic acid will be in the gas phase at pH 4 and 5 under cloud water conditions, whereas all of oxalic acid, malonic acid, and maleic acid will be in the aqueous phase at pH 4 and 5 under cloud water conditions. This suggests that gas-phase chemical reactions will likely play an important role in consuming formic acid, whereas the consumption of oxalic acid, malonic acid, and maleic acid will likely mainly be through bacterial activity and chemical reactions in the aqueous phase. Quantifying the exact contributions of aqueous-phase bacterial activity vs. aqueous-phase ·OH/NO₃· chemistry vs. gas-phase ·OH/NO₃· chemistry under different cloud water pH conditions will require a multi-phase box model similar to the one used by Khaled et al. (2021). This is beyond the scope of the current study but can be a subject of future studies."

8. Referee comment: "*L46: what are "microbiological-ecosystem interactions"? these are likely not precisely atmospheric processes.*"

**Author response:** This phrase has been removed from the sentence in the revised manuscript.

9. Referee comment: "*L54: why "however"? there is no contradiction here.*"

**Author response:** This word has been removed in the revised manuscript.

10. Referee comment: "*L57: "The cell concentrations of metabolically active bacteria in clouds typically range from about $10^2$ to $10^5$ cells $mL^{-1}$": these are numbers for the concentration of total bacteria, not only those metabolically active.*"

**Author response:** We have removed the phrase "metabolically active" from this sentence in the revised manuscript.

11. Referee comment: "*L60-61: The authors should read the references that they cite, by far not only culturable bacteria have been investigated in clouds: (Amato et al., 2017, 2019) rely on molecular data (rRNA and rDNA), and reference could also be made to (Péguilhan et al., 2021).*"

**Author response:** We have revised this statement in the revised manuscript:

**Page 2, line 66: "At present, our knowledge on bacterial communities in clouds are limited to the few areas that have been studied (e.g., Puy de Dôme in France, Mt. Tai in North China) (Amato et al., 2005; Amato et al., 2017; Wei et al., 2017; Péguilhan et al., 2021). Cultural bacteria typically makes up a very small fraction (about 1%) of the entire**

**bacteria community in clouds (Amato et al., 2005).”**

12. Referee comment: “*L66: (Zhang et al., 2021) does not investigate microbiological-chemical interactions but physical impacts of bioaerosols.*”

**Author response:** We have removed the reference.

13. Referee comment: “*L67: “Many bacteria species isolated from cloud water have the enzymes needed to biodegrade organic compounds such as carboxylic acids, formaldehyde, methanol, phenolic compounds, and amino acids”: is this specific of bacteria in clouds? All bacteria carry at least some of the functions listed so this is a bit misleading.*”

**Author response:** We have rewritten this sentence in the revised manuscript:

**Page 3, line 84: “Many bacteria species have the enzymes needed to biodegrade organic compounds. Some of the bacteria species isolated from cloud water could biodegrade organic acids, formaldehyde, methanol, phenolic compounds, and amino acids (Ariya et al., 2002; Husárová et al., 2011; Vaïtilingom et al., 2011; Jaber et al., 2020; Jaber et al., 2021).”**

14. Referee comment: “*L70: “the bacteria need to be metabolically active to biodegrade organic compounds” is a Pleonasm*”

**Author response:** This sentence has been removed from the revised manuscript.

15. Referee comment: “*L74: “mimicking the Puy de Dôme” è “mimicking cloud water chemical composition at puy de Dôme””*

**Author response:** We have made the requested changes in the revised manuscript.

16. Referee comment: “*L90: “influence”: the fact that a link was found between these parameters does not imply causal relationship.*”

**Author response:** We have replaced the word “influences” with the word “impacts” in the revised manuscript.

17. Referee comment: “*L126: “suggested”? is the appropriate word? (Isn’t the isolation of strains factual?)*”

**Author response:** We have replaced the word “suggested” with the word “reported” in the revised manuscript.

18. *L128:* Referee comment: “*Enterobacter is pathogenic”. Are all Enterobacter pathogenic? Have these strains in particular been tested?*”

**Author response:** This sentence has been removed from the revised manuscript.

19. Referee comment: “*L 142: what sampling method was used? Give more detail about the isolation of these strains.*”

**Author response:** We have added the following information to the revised manuscript:

**Page 5, line 167: "Two new strains (B0910 and pf0910) belonging to *Enterobacter* species were isolated by exposing nutrient agar plates to ambient air in an urban environment (22.3360° N, 114.1732° E) at the height of 50 m above sea level during the summer season (~22 °C) in Hong Kong."**

20. Referee comment: "*L137, 214 and elsewhere: MSA in not a carboxylic acid. Use relevant terms. In addition, provide a reference for its concentration in cloud water (I did not find mention of MSA in the papers cited).*"

**Author response:** We have replaced "carboxylic acid(s)" with "organic acid(s)" in the revised manuscript. One of the references we cited in the manuscript (Sun et al., 2016) reported that MSA concentrations are in range of 0.1 to 3.41 μM.

21. Referee comment: "*L238: "Control experiments were performed using solutions that contained carboxylic acids but no bacterial cells.": were this carried out for both light and dark conditions?*"

**Author response:** Control experiments were conducted under both light and dark conditions. This information has been added to the revised manuscript:

**Page 9, line 299: "Control experiments were performed under illuminated and dark conditions using solutions that contained organic acids but no bacterial cells. The organic acids did not degrade in these control experiments."**

22. Referee comment: "*L247: "with normal functioning cells usually maintaining a constant ADP/ATP ratio": what is normal functioning? And this statement requires a reference.*"

**Author response:** We meant "growing cells", not "normal functioning cells". We refer the referee to our response to comment 23.

23. Referee comment: "*L249: Specify what are ADP and ATP molecules, why these are important in biological systems, and how their ratio relates with metabolic activity.*"

**Author response:** We have added the following information to the revised manuscript:

**Page 7, line 225: "The adenosine diphosphate/adenosine triphosphate (ADP/ATP) ratios were measured using an assay kit (EnzyLightTM, BioAssay Systems) and a bioluminteter (SpectraMax M2e) to determine changes in the bacteria energetic metabolism."**

**Page 9, line 305: "Figure 1 shows the survival rates and ADP/ATP ratios of the *E. hormaechei* B0910 and *E. hormaechei* pf0910 strains over time under illuminated and dark conditions at different artificial cloud water pH. The ADP/ATP ratio is used an indicator of the bacteria's metabolic activity and survival rate in this study. Growing cells usually maintain a constant ADP/ATP ratio because whenever there is a decrease in intracellular ATP production, its degradation product ADP will be resynthesized to form ATP to maintain intracellular ATP concentrations (Koutny et al., 2006; Guan and Liu, 2020). In contrast, when there is a disruption in the metabolism of ATP production, ATP**

**cannot be resynthesized from ADP even though ATP is still converted to ADP, which will cause the ADP/ATP ratio to increase (Koutny et al., 2006; Guan and Liu, 2020)."**

24. Referee comment: "*L281: "Both strains will likely not survive in pH < 4 cloud water during the daytime and nighttime". There should be a mention to time here. What is the timescale considered? What is the half-life time at pH > 4 versus < 4?*"

**Author response:** We have added the following information to the revised manuscript:

**Page 12, line 361: "Based on our results, we estimate that the half-lives of the bacteria strains in pH 4.3 cloud water under illumination conditions (e.g., light intensity, wavelengths) similar to those in our study are around 430 min. The half-lives of the bacteria strains in pH < 4 are cloud water are lower. Based on our results, we estimate that the daytime and nighttime half-lives of the bacteria strains in pH 3.3 cloud water are around 2 min."**

25. *L295: not "production" of compounds, but "release", since these are from cell lysis. Figure 3: Why have 2 distinct X axes? They seem to correspond directly to each other.*

**Author response:** We have replaced the word "production" with the word "release" in the revised manuscript. The lower x axes in Figure 3 are used for the stacked bar chart, which shows the time evolution of the UPLC-MS TIC signals of the classes of compounds released at five time points. The upper x axes are used for the time evolution of the survival rates and ADP/ATP ratios at twelve time points. We found having a single x axis made it difficult to display the results clearly since the time points at which samples were taken for UPLC-MS measurements were different from those taken for survival rates determination and ADP/ATP measurements. Thus, we decided to use two distinct x axes to display the above-mentioned time evolutions clearly.

26. *L351-359: It would be interesting to discuss more about the discrepancy between the compounds released by cells during lysis and their composition: what are the proportions expected of the different categories of compounds (i.e. what are these proportions in the cellular material)?*

**Author response:** As requested, we have added more discussion to the revised manuscript:

**Page 16, line 429: "To better understand the compounds released by the two strains, the O/C and H/C elemental ratios of the identified compounds were used to construct Van Krevelen (VK) diagrams. Regions of the VK diagrams were assigned to eight chemical classes based on the combined O/C and H/C ratios: lipids, unsaturated hydrocarbons, condensed aromatic structures, peptides, lignin, tannin, amino sugars, and carbohydrates (Table S4) (Bianco et al., 2018; Laszakovits and Mackay, 2022). Rivas-Ubach et al. (2018) previously reported that the region of the VK diagram assigned to amino sugars overlaps with the region for nucleic acids. Figures S8 and S9 show the VK diagrams for water-soluble and water-insoluble compounds released by *E. hormaechei* B0910, respectively, while Figures S10 and S11 show the VK diagrams for water-soluble and water-insoluble compounds released by *E. hormaechei* pf0910, respectively. Majority of the water-soluble**

and water-insoluble compounds released from both strains (50% to 60%) were assigned as lipids based on their O/C and H/C ratios, while the second most abundant compound class was peptides (10% to 20%). The two least abundant compound classes were amino sugars/nucleic acids and carbohydrates. Since the dry matter of a typical bacterial cell contains approximately 55% proteins and amino acids, 24% nucleic acids, 10% carbohydrates, 7% lipids, and 5% inorganic minerals and trace elements (Watson et al., 2007), the differences in the abundance of compound classes detected vs. the dry matter of a typical bacterial cell indicated that cellular components were likely biologically and/or chemically modified during and after cell lysis during exposure to light. For instance, the large abundance of peptides detected could be a result of biological and/or chemical modifications of proteins and amino acids, which comprise majority of the dry matter of a typical bacterial cell. Peptide bonds are formed by biochemical reactions where a water molecule is removed as the amino group of one amino acid is joined to the carboxyl group of a neighboring amino acid. The large abundance of lipids was unsurprising since lipids are the main component of cell membranes so large quantities of lipids are expected from the lysed cells. Most of the lipid molecules released during cell lysis may not have undergone biological and/or chemical modifications under our experimental conditions. The two least abundant compound classes were amino sugars/nucleic acids and carbohydrates. This was somewhat surprising since nucleic acids and carbohydrates are abundant in the dry matter of a typical bacterial cell. It is possible that these compounds were biologically and/or chemically modified to form other compounds (e.g., exopolymeric substances) during exposure to light (Matulova et al., 2014). In addition, the extraction procedure employed (Section S2) may not have extracted these compounds effectively for analysis. For instance, nucleic acids and carbohydrates are polar molecules, which are difficult to retain on the solid phase extraction columns used in this study. These compounds may also have been poorly separated in UPLC and/or inefficiently ionized by ESI."

27. Referee comment: "*L365: Again, MSA in not a carboxylic acid.*"

**Author response:** We have replaced "carboxylic acid(s)" with "organic acid(s)" in the revised manuscript.

28. Referee comment: "*L463-482: this not a conclusion but a summary.*"

**Author response:** We have replaced the word "conclusion" with the word "summary" in the revised manuscript.

29. *L 487: Results from this study imply that there is a minimum cloud water pH threshold at which the bacteria will survive and thrive in during the daytime and/or nighttime". What is that threshold? The data does not show the existence of a threshold per se, so this is a bit overstated.*

**Author response:** We have revised this sentence in the revised manuscript:

**Page 27, line 742: "Results from this study imply the cloud water pH will impact the bacteria's ability to survive and thrive in during the daytime and/or nighttime."**

**Reference:**

Amato, P., Besaury, L., Joly, M., Penaud, B., Deguillaume, L., and Delort, A.-M.: Metatranscriptomic exploration of microbial functioning in clouds, Scientific Reports, 9, 1-12, https://doi.org/10.1038/s41598-019-41032-4, 2019.

Amato, P., Ménager, M., Sancelme, M., Laj, P., Mailhot, G., and Delort, A.-M.: Microbial population in cloud water at the Puy de Dôme: Implications for the chemistry of clouds, Atmospheric Environment, 39, 4143-4153, https://doi.org/10.1016/j.atmosenv.2005.04.002, 2005.

Amato, P., Parazols, M., Sancelme, M., Mailhot, G., Laj, P., and Delort, A.-M.: An important oceanic source of micro-organisms for cloud water at the Puy de Dôme (France), Atmospheric Environment, 41, 8253-8263, https://doi.org/10.1016/j.atmosenv.2007.06.022, 2007.

Amato, P., Joly, M., Besaury, L., Oudart, A., Taib, N., Mone, A. I., Deguillaume, L., Delort, A. M., and Debroas, D.: Active microorganisms thrive among extremely diverse communities in cloud water, PLoS One, 12, e0182869, https://doi.org/10.1371/journal.pone.0182869, 2017.

Ariya, P. A., Nepotchatykh, O., Ignatova, O., and Amyot, M.: Microbiological degradation of atmospheric organic compounds, Geophysical Research Letters, 29, 34-31-34-34, https://doi.org/10.1029/2002gl015637, 2002.

Bianco, A., Voyard, G., Deguillaume, L., Mailhot, G., and Brigante, M.: Improving the characterization of dissolved organic carbon in cloud water: Amino acids and their impact on the oxidant capacity, Sci Rep, 6, 37420, https://doi.org/10.1038/srep37420, 2016.

Bianco, A., Deguillaume, L., Chaumerliac, N., Vaïtilingom, M., Wang, M., Delort, A.-M., and Bridoux, M. C.: Effect of endogenous microbiota on the molecular composition of cloud water: a study by Fourier-transform ion cyclotron resonance mass spectrometry (FT-ICR MS), Scientific Reports, 9, 7663, https://doi.org/10.1038/s41598-019-44149-8, 2019.

Bianco, A., Deguillaume, L., Vaitilingom, M., Nicol, E., Baray, J. L., Chaumerliac, N., and Bridoux, M.: Molecular Characterization of Cloud Water Samples Collected at the Puy de Dome (France) by Fourier Transform Ion Cyclotron Resonance Mass Spectrometry, Environmental Science & Technology, 52, 10275-10285, https://doi.org/10.1021/acs.est.8b01964, 2018.

Brzoska, R. M., Edelmann, R. E., and Bollmann, A.: Physiological and Genomic Characterization of Two Novel Bacteroidota Strains Asinibacterium spp. OR43 and OR53, Bacteria, 1, 33-47, 2022.

Clegg, S. L., Brimblecombe, P., and Khan, L.: The Henry's law constant of oxalic acid and its partitioning into the atmospheric aerosol, Idojaras, 100, 51-68, 1996.

Compernolle, S. and Müller, J. F.: Henry's law constants of diacids and hydroxy polyacids: recommended values, Atmos. Chem. Phys., 14, 2699-2712, https://doi.org/10.5194/acp-14-2699-2014, 2014.

Delort, A.-M., Vaïtilingom, M., Amato, P., Sancelme, M., Parazols, M., Mailhot, G., Laj, P., and Deguillaume, L.: A short overview of the microbial population in clouds: Potential roles in atmospheric chemistry and nucleation processes, Atmospheric Research, 98, 249-260, https://doi.org/10.1016/j.atmosres.2010.07.004, 2010.

Ervens, B. and Amato, P.: The global impact of bacterial processes on carbon mass, Atmospheric Chemistry and Physics, 20, 1777-1794, https://doi.org/10.5194/acp-20-1777-2020, 2020.

Ervens, B., Turpin, B. J., and Weber, R. J.: Secondary organic aerosol formation in cloud droplets and aqueous particles (aqSOA): a review of laboratory, field and model studies, Atmos. Chem. Phys., 11, 11069-11102, 10.5194/acp-11-11069-2011, 2011.

Fankhauser, A. M., Antonio, D. D., Krell, A., Alston, S. J., Banta, S., and McNeill, V. F.: Constraining the Impact of Bacteria on the Aqueous Atmospheric Chemistry of Small Organic Compounds, ACS Earth and Space Chemistry, 3, 1485-1491, https://doi.org/10.1021/acsearthspacechem.9b00054, 2019.

Guan, N. and Liu, L.: Microbial response to acid stress: mechanisms and applications, Applied Microbiology and Biotechnology, 104, 51-65, https://doi.org/10.1007/s00253-019-10226-1, 2020.

Guo, H., Liu, J., Froyd, K. D., Roberts, J. M., Veres, P. R., Hayes, P. L., Jimenez, J. L., Nenes, A., and Weber, R. J.: Fine particle pH and gas–particle phase partitioning of inorganic species in Pasadena, California, during the 2010 CalNex campaign, Atmos. Chem. Phys., 17, 5703-5719, https://doi.org/10.5194/acp-17-5703-2017, 2017.

Guo, H., Sullivan, A. P., Campuzano-Jost, P., Schroder, J. C., Lopez-Hilfiker, F. D., Dibb, J. E., Jimenez, J. L., Thornton, J. A., Brown, S. S., Nenes, A., and Weber, R. J.: Fine particle pH and the partitioning of nitric acid during winter in the northeastern United States, Journal of Geophysical Research: Atmospheres, 121, 10,355-310,376, https://doi.org/10.1002/2016JD025311, 2016.

Haynes, W. M.: CRC handbook of chemistry and physics, CRC Press, Boca Raton, Florida2014.

Herrmann, H., Hoffmann, D., Schaefer, T., Brauer, P., and Tilgner, A.: Tropospheric aqueous-phase free-radical chemistry: radical sources, spectra, reaction kinetics and prediction tools, Chemphyschem, 11, 3796-3822, https://doi.org/10.1002/cphc.201000533, 2010.

Husárová, S., Vaïtilingom, M., Deguillaume, L., Traikia, M., Vinatier, V., Sancelme, M., Amato, P., Matulová, M., and Delort, A.-M.: Biotransformation of methanol and formaldehyde by bacteria isolated from clouds. Comparison with radical chemistry, Atmospheric Environment, 45, 6093-6102, https://doi.org/10.1016/j.atmosenv.2011.06.035, 2011.

Jaber, S., Joly, M., Brissy, M., Leremboure, M., Khaled, A., Ervens, B., and Delort, A.-M.: Biotic and abiotic transformation of amino acids in cloud water: experimental studies and atmospheric implications, Biogeosciences, 18, 1067-1080, https://doi.org/10.5194/bg-18-1067-2021, 2021.

Jaber, S., Lallement, A., Sancelme, M., Leremboure, M., Mailhot, G., Ervens, B., and Delort, A.-M.: Biodegradation of phenol and catechol in cloud water: comparison to chemical oxidation in the atmospheric multiphase system, Atmospheric Chemistry and Physics, 20, 4987-4997, https://doi.org/10.5194/acp-20-4987-2020, 2020.

Kawamura, K., Ishimura, Y., and Yamazaki, K.: Four years' observations of terrestrial lipid class compounds in marine aerosols from the western North Pacific, Global Biogeochemical Cycles, 17, https://doi.org/10.1029/2001gb001810, 2003.

Khaled, A., Zhang, M., Amato, P., Delort, A.-M., and Ervens, B.: Biodegradation by bacteria in clouds: an underestimated sink for some organics in the atmospheric multiphase system, Atmospheric Chemistry and Physics, 21, 3123-3141, https://doi.org/10.5194/acp-21-3123-2021, 2021.

Koutny, M., Sancelme, M., Dabin, C., Pichon, N., Delort, A.-M., and Lemaire, J.: Acquired biodegradability of polyethylenes containing pro-oxidant additives, Polymer Degradation and Stability, 91, 1495-1503, https://doi.org/10.1016/j.polymdegradstab.2005.10.007, 2006.

Krulwich, T. A., Sachs, G., and Padan, E.: Molecular aspects of bacterial pH sensing and homeostasis, Nature Reviews Microbiology, 9, 330-343, 2011.

Krumins, V., Mainelis, G., Kerkhof, L. J., and Fennell, D. E.: Substrate-dependent rRNA production in an airborne bacterium, Environmental Science & Technology Letters, 1, 376-381, https://doi.org/10.1021/ez500245y, 2014.

Laszakovits, J. R. and MacKay, A. A.: Data-Based Chemical Class Regions for Van Krevelen Diagrams, Journal of the American Society for Mass Spectrometry, 33, 198-202, https://doi.org/10.1021/jasms.1c00230, 2022.

Lee, A. K. Y., Chan, C. K., Fang, M., and Lau, A. P. S.: The 3-hydroxy fatty acids as biomarkers for quantification and characterization of endotoxins and Gram-negative bacteria in atmospheric aerosols in Hong Kong, Atmospheric Environment, 38, 6307-6317, https://doi.org/10.1016/j.atmosenv.2004.08.013, 2004.

Lide, D. R. and Frederikse, H. P. R.: CRC handbook of chemistry and physics, Boca Raton, Florida1995.

Matulova, M., Husarova, S., Capek, P., Sancelme, M., and Delort, A. M.: Biotransformation of various saccharides and production of exopolymeric substances by cloud-borne Bacillus sp. 3B6, Environ Sci Technol, 48, 14238-14247, https://doi.org/10.1021/es501350s, 2014.

Meskhidze, N., Chameides, W. L., Nenes, A., and Chen, G.: Iron mobilization in mineral dust:

Can anthropogenic SO2 emissions affect ocean productivity?, Geophysical Research Letters, 30, https://doi.org/10.1029/2003GL018035, 2003.

Nah, T., Guo, H., Sullivan, A. P., Chen, Y., Tanner, D. J., Nenes, A., Russell, A., Ng, N. L., Huey, L. G., and Weber, R. J.: Characterization of aerosol composition, aerosol acidity, and organic acid partitioning at an agriculturally intensive rural southeastern US site, Atmos. Chem. Phys., 18, 11471-11491, https://doi.org/10.5194/acp-18-11471-2018, 2018.

Péguilhan, R., Besaury, L., Rossi, F., Enault, F., Baray, J.-L., Deguillaume, L., and Amato, P.: Rainfalls sprinkle cloud bacterial diversity while scavenging biomass, FEMS Microbiology Ecology, 97, https://doi.org/10.1093/femsec/fiab144, 2021.

Rivas-Ubach, A., Liu, Y., Bianchi, T. S., Tolic, N., Jansson, C., and Pasa-Tolic, L.: Moving beyond the van Krevelen diagram: A new stoichiometric approach for compound classification in organisms, Analytical chemistry, 90, 6152-6160, https://doi.org/10.1021/acs.analchem.8b00529, 2018.

Sander, R.: Compilation of Henry's law constants (version 4.0) for water as solvent, Atmos. Chem. Phys., 15, 4399-4981, https://doi.org/10.5194/acp-15-4399-2015, 2015.

Saxena, P. and Hildemann, L. M.: Water-soluble organics in atmospheric particles: A critical review of the literature and application of thermodynamics to identify candidate compounds, Journal of Atmospheric Chemistry, 24, 57-109, https://doi.org/10.1007/BF00053823, 1996.

Sun, X., Wang, Y., Li, H., Yang, X., Sun, L., Wang, X., Wang, T., and Wang, W.: Organic acids in cloud water and rainwater at a mountain site in acid rain areas of South China, Environ Sci Pollut Res Int, 23, 9529-9539, https://doi.org/10.1007/s11356-016-6038-1, 2016.

Tyagi, P., Ishimura, Y., and Kawamura, K.: Hydroxy fatty acids in marine aerosols as microbial tracers: 4-year study on β-and ω-hydroxy fatty acids from remote Chichijima Island in the western North Pacific, Atmospheric Environment, 115, 89-100, 2015.

Vaitilingom, M., Amato, P., Sancelme, M., Laj, P., Leriche, M., and Delort, A. M.: Contribution of microbial activity to carbon chemistry in clouds, Appl Environ Microbiol, 76, 23-29, https://doi.org/10.1128/AEM.01127-09, 2010.

Vaitilingom, M., Deguillaume, L., Vinatier, V., Sancelme, M., Amato, P., Chaumerliac, N., and Delort, A. M.: Potential impact of microbial activity on the oxidant capacity and organic carbon budget in clouds, Proc Natl Acad Sci U S A, 110, 559-564, https://doi.org/10.1073/pnas.1205743110, 2013.

Vaïtilingom, M., Charbouillot, T., Deguillaume, L., Maisonobe, R., Parazols, M., Amato, P., Sancelme, M., and Delort, A. M.: Atmospheric chemistry of carboxylic acids: microbial implication versus photochemistry, Atmospheric Chemistry and Physics, 11, 8721-8733, https://doi.org/10.5194/acp-11-8721-2011, 2011.

Watson, J., Baker, T., and Bell, S.: Molecular biology of the gene, 6th edn. W,   2007.

Weast, R. C. and Astle, M. J.: CRC Handbook of Chemistry and Physics, CRC Press1981.

Wei, M., Xu, C., Chen, J., Zhu, C., Li, J., and Lv, G.: Characteristics of bacterial community in cloud water at Mt Tai: similarity and disparity under polluted and non-polluted cloud episodes, Atmos. Chem. Phys., 17, 5253-5270, https://doi.org/10.5194/acp-17-5253-2017, 2017.

pKa data compiled by R. Williams: https://organicchemistrydata.org/hansreich/resources/pka/pka_data/pka-compilation-williams.pdf, last access: 7 Dec 2022.

---

## Editor Decision (ED1)

**Final decision-acp-2022-608**

Based on the detailed comments of the two experts in the field, and after my consideration, the manuscript is of adequate atmospheric interest to merit publication in *Atmospheric Chemistry and Physics.*

The authors have thoroughly responded all the questions/comments raised by the reviewers, and modified the manuscript according to the suggestions. Besides, the revised manuscript has been checked again by Pierre Amato, who now supports the publication.

The manuscript can be accepted for publication in ACP.

**Additional comments:**

Can you please add the aliquots of the sample removed for analysis where necessary?

See Referee 1 comment:

*"These experiments were conducted in 5 mL volumes based on Methods section 2.2 (pg 6, ln 176). Multiple points in the manuscript (e.g., pg 6, ln 179; pg 7, ln 196; pg 8, ln 226) mentioned aliquots of the sample being removed for analysis....."*

Sincerely,

Irena Grgić

---

## Author Response (AR2)

**Referee comment:** Can you please add the aliquots of the sample removed for analysis where necessary?

See Referee 1 comment:

"These experiments were conducted in 5 mL volumes based on Methods section 2.2 (pg 6, ln 176). Multiple points in the manuscript (e.g., pg 6, ln 179; pg 7, ln 196; pg 8, ln 226) mentioned aliquots of the sample being removed for analysis….."

**Author response:** We have made the requested additions to the revised manuscript:

**Page 7 line 184: "100 µL of sample were removed at different time points for Colony Forming Unit (CFU) counts on LB agar at 37 °C for 16 hours to determine the culturable bacterial cell concentrations, which was used to calculate the bacteria survival rates. 20 µL of sample were removed at different time points for measurements of adenosine diphosphate/adenosine triphosphate (ADP/ATP) ratios using an assay kit (EnzyLight™, BioAssay Systems) and a biolumineter (SpectraMax M2e) to determine changes in the bacteria energetic metabolism."**

**Page 7 line 201: "During some experiments, aliquots of the solutions (10 mL) were taken at time points 0 h, 2 h, 4 h, 8 h, and 12 h and analyzed by ultra-performance liquid chromatography-mass spectrometry (UPLC-MS)."**

**Page 9 line 240: "During each experiment, aliquots of the solutions (0.6 mL) were taken every 2 hours over 12 hours. The organic acid concentrations in each filtered aliquot of solution were measured by ion chromatography (IC) using a Dionex ICS-1100 (ThermoFisher Scientific) system."**